# Modified K-means Algorithm with Local Optimality Guarantees

**Mingyi Li** [1]  **Michael R. Metel** [2]  **Akiko Takeda** [1 3]

## Abstract

The K-means algorithm is one of the most widely studied clustering algorithms in machine learning. While extensive research has focused on its ability to achieve a globally optimal solution, there still lacks a rigorous analysis of its local optimality guarantees. In this paper, we first present conditions under which the K-means algorithm converges to a locally optimal solution. Based on this, we propose simple modifications to the K-means algorithm which ensure local optimality in both the continuous and discrete sense, with the same computational complexity as the original K-means algorithm. As the dissimilarity measure, we consider a general Bregman divergence, which is an extension of the squared Euclidean distance often used in the K-means algorithm. Numerical experiments confirm that the K-means algorithm does not always find a locally optimal solution in practice, while our proposed methods provide improved locally optimal solutions with reduced clustering loss. Our code is available at https://github.com/lmingyi/LO-K-means.

## 1. Introduction

K-means clustering is one of the most widely recognized methods for partitioning datasets into homogeneous groups. The K-means clustering problem, formulated as a nonconvex optimization problem in (P1), is NP-hard (Aloise et al., 2009). The K-means algorithm, also known as Lloyd's algorithm, was first proposed by Lloyd (1982) as a quantization method for pulse-code modulation and has since been widely applied and extensively studied across various domains, including image recognition, text mining, deep learning model weight quantization, healthcare, mar-

keting, and education (Csurka et al., 2004; Steinbach et al., 2000; Han et al., 2016; Kim et al., 2024; Liu & Shih, 2005; Bharara et al., 2018; Moubayed et al., 2020).

It is well known that the solution generated by the K-means algorithm is highly sensitive to the algorithm's initialization. To mitigate this issue, various strategies have been introduced. These include methods for improving the initial cluster centers, such as K-means++ (Arthur & Vassilvitskii, 2006), and approaches that involve running the algorithm multiple times with different initializations (Jain, 2010). Nevertheless, it still seems to be largely accepted from at least the 1980s that the K-means algorithm converges to a locally optimal solution: the scikit-learn documentation states that "in practice, the K-means algorithm is very fast... but it falls in local minima" (scikit-learn developers, 2025), while Grunau & Rozhoň (2022) confirm that "Lloyd's algorithm converges only to a local optimum". Similarly, Balcan et al. (2018) describe that "Lloyd's method is used to converge to a local minimum". For the local optimality guarantee, (Selim & Ismail, 1984) is often cited, which attempted to prove that the K-means algorithm converges to a locally optimal solution, but their proof only demonstrated that the K-means algorithm converges within a finite number of iterations, and indeed, we show by counterexample that the K-means algorithm does not always converge to a locally optimal solution in Section 3.1.

This prevalent assumption about local optimality has also shaped research on improving the K-means algorithm. Variants such as K-means--, which is robust to outliers, and K-means clustering using Bregman divergences, are also claimed to converge to a locally optimal solution (Chawla & Gionis, 2013, Theorem 4.1, Banerjee et al., 2005, Proposition 3). In these studies, the convergence to a locally optimal solution is justified by the fact that the loss function monotonically decreases after each iteration. However, there still appears to be a gap between these proofs and a guarantee of local optimality. Based on this, the following question naturally arises:

*Can we modify the K-means algorithm to guarantee local optimality while maintaining the same order of computational complexity?*

The answer is "yes" for the K-means algorithm using

---

[1]Department of Mathematical Informatics, The University of Tokyo, Tokyo, Japan [2]Huawei Noah's Ark Lab, Montréal, Canada [3]Center for Advanced Intelligence Project, RIKEN, Tokyo, Japan. Correspondence to: Michael R. Metel <michael.metel@h-partners.com>.

*Proceedings of the $42^{nd}$ International Conference on Machine Learning*, Vancouver, Canada. PMLR 267, 2025. Copyright 2025 by the author(s).

the squared Euclidean distance, as well as the more general setting of weighted K-means using Bregman divergences, which we focus on in this work.

Besides (Selim & Ismail, 1984), another paper related to our work is (Peng & Xia, 2005), which considers the K-means problem with squared Euclidean distance, and gives necessary and sufficient conditions for locally optimal solutions. An algorithm which converges to a D-local optimal solution is presented (see Definition 2.3), as well as a method to find a global optimal solution. Recognizing the dominance of the K-means algorithm for solving the K-means problem, our focus was not to create a completely new algorithm, but to develop simple modifications to the K-means algorithm which can be easily added into already established codebases. With this goal in mind, our necessary and sufficient conditions for local optimality, given in Section 4, were developed for verification of solutions generated by the K-means algorithm, while also being applicable for general Bregman divergences.

**Our contribution**

- The K-means algorithm has long been assumed to converge to a locally optimal solution, even if it does not reach a globally optimal solution. To the best of our knowledge, this is the first study to point out this misconception. We provide a counterexample in Section 3.1 showing that the K-means algorithm does not always converge to a locally optimal solution.

- We propose practical modifications to the K-means algorithm that introduce minimal computational and implementation overhead. It is proven that these modifications guarantee that the K-means algorithm converges to either a locally optimal solution of its continuous relaxation (called C-local, see Definition 2.4) or to a discrete locally optimal solution (D-local), without increasing the per-iteration time complexity nor space complexity of the original K-means algorithm.

- By performing extensive experiments using both synthetic and real-world datasets using different Bregman divergences, we find that our modified K-means algorithms (LO-K-means) result in improved solutions with decreased clustering loss. In particular, one of our proposed variants, Min-D-LO, seems to strike the correct balance between improved accuracy and computation time, by being able to significantly decrease the clustering loss, while also being much faster than other tested D-local algorithms, including the algorithm proposed in (Peng & Xia, 2005, Section 3.2.1).

**Notation** For an $N \in \mathbb{N}$, let $[N] := \{1, 2, \ldots, N\}$. Let $\mathbb{R}$, $\mathbb{R}_+$, and $\mathbb{R}_{++}$ denote the set of real numbers, nonnegative real numbers, and positive real numbers, respectively. The effective domain of a function $f : \mathbb{R}^d \to \mathbb{R} \cup \{+\infty\}$ is defined as $\mathrm{dom}(f) := \{x \in \mathbb{R}^d \mid f(x) < +\infty\}$. For a set $X \subset \mathbb{R}^d$, let $\mathrm{co}(X)$ denote its convex hull. In a Euclidean metric space $(X, d)$, let $B(a, \epsilon) := \{x \in X \mid d(a, x) < \epsilon\}$ represent an open ball of radius $\epsilon > 0$ centered at $a \in X$. To describe the rate of convergence, we employ Landau notation. For functions $f$ and $g : \mathbb{R}_+ \to \mathbb{R}_+$, the notation $f(x) = O(g(x))$ is used as $x \to \infty$ to indicate that there exists a constant $M > 0$ such that $|f(x)| < M|g(x)|$ for sufficiently large $x$.

## 2. Preliminaries

### 2.1. Problem Formulation

In this work, we consider a generalized K-means clustering problem, the weighted K-means problem using Bregman divergences, where we are given a set of $N$ unique points in $\mathbb{R}^d$, $X = \{x_i\}_{i=1}^N$, with corresponding positive weights $W = \{w_i\}_{i=1}^N \subset \mathbb{R}_{++}$, and a number of clusters $K \in \mathbb{N}$ to partition the $N$ points of $X$.

We use $P \in \mathbb{R}^{K \times N}$ to denote the assignment of points to clusters, where $p_{i,j} \in \{0, 1\}$, and $p_{i,j} = 1$ indicates that point $j$ is assigned to cluster $i$. The centers of the clusters are represented by $C \in \mathbb{R}^{d \times K}$, where each column corresponds to a center $c_i \in \mathbb{R}^d$ for $i = 1, 2, \ldots, K$. Furthermore, the sum of the weights of points belonging to each cluster $k$ is expressed as $s_k(P) = \sum_{n=1}^N p_{k,n} w_n$. We denote the dissimilarity measure as $D(x, c)$, which we assume is a Bregman divergence (see Section 2.2, e.g. $D(x, c) = \|x - c\|_2^2$ for the squared Euclidean distance).

The weighted K-means problem is formulated as follows.

$$
\text{(P1)} \quad \min_{P, C} \quad f(P, C) = \sum_{k=1}^K \sum_{n=1}^N p_{k,n} w_n D(x_n, c_k)
$$
$$
\text{s.t.} \quad P \in S_1, \ c_k \in R \quad \forall k \in [K],
$$

where the feasible region $S_1$ of $P$ is given as

$$
\left\{ P \in \mathbb{R}^{K \times N} \ \middle| \ \begin{array}{ll} \sum_{k=1}^K p_{k,n} = 1 & \forall n \in [N], \\ p_{k,n} \in \{0, 1\} & \forall k \in [K], n \in [N] \end{array} \right\} \tag{1}
$$

and the feasible region $R \subseteq \mathbb{R}^d$ of all $c_k$ is dependent on the choice of $D$ (when $D(x, c) = \|x - c\|_2^2$, $R = \mathbb{R}^d$). The first constraint in (1) ensures that each point is assigned to one cluster. Throughout this paper, we refer to $f(P, C)$ as the clustering loss.

If the number of points in the dataset, $N$, is less than or equal to the number of clusters $K$, the global optimal value

of (P1) is trivially equal to $0$. Therefore, this work focuses on situations where $N > K$. The assumed uniqueness of all $x \in X$ is also without a loss of generality, given that if there exists $x_i, x_j \in X$ such that $x_i = x_j$, the point $x_j$ can be removed from $X$, with $w_i$ updated to equal $w_i + w_j$.

We also consider the following problem, where the domain $S_2$ of $P$ is a continuous relaxation of $S_1$ in (P1):

$$(\text{P2}) \quad \min_{P,C} \quad f(P, C) = \sum_{k=1}^{K} \sum_{n=1}^{N} p_{k,n} w_n D(x_n, c_k)$$
$$\text{s.t.} \quad P \in S_2, \; c_k \in R \quad \forall k \in [K],$$

where

$$S_2 := \left\{ P \in \mathbb{R}^{K \times N} \; \middle| \; \begin{array}{l} \sum_{k=1}^{K} p_{k,n} = 1 \quad \forall n \in [N], \\ p_{k,n} \geq 0 \quad \forall k \in [K], n \in [N] \end{array} \right\}.$$

It is noted that $S_1$ is the set of all extreme points of $S_2$.

For a given $P \in S_2$, we define

$$F(P) := \min_{C \in R^K} f(P, C), \qquad (2)$$
$$C^*(P) := \arg\min_{C \in R^K} f(P, C).$$

In general $C^*(P)$ is not a singleton. We denote an element of $C^*(P)$ as $C_P^*$, i.e., $C_P^* \in C^*(P)$, and denote its $k$-th column (cluster center) as $(c_P^*)_k$.

We can rewrite the optimization problem (P1) as simply $\min_{P \in S_1} F(P)$, and similarly for $(P2)$. It is known (Selim & Ismail, 1984, Lemma 1) that $F(P)$ is concave, which is easily derived from the linearity of the clustering loss with respect to $P$, irrespective of the choice of $D$.

In addition, we define $A(C)$ as the set of optimal extreme points that minimize the clustering loss when $C$ is fixed,

$$A(C) := \arg\min_{P \in S_1} f(P, C). \qquad (3)$$

### 2.2. Bregman Divergences

In this work we focus on dissimilarity measures $D$ belonging to the class of Bregman divergences, which generalize the squared Euclidean distance.

**Definition 2.1** (Bregman (1967))**.** Let $\phi : \text{dom}(\phi) \to \mathbb{R}$ be a strictly convex function defined on a convex set $\text{dom}(\phi)$ such that $\phi$ is differentiable on $\text{int dom}(\phi)$. The Bregman divergence $D_\phi : \text{dom}(\phi) \times \text{int dom}(\phi) \to \mathbb{R}_+$ is defined as

$$D_\phi(x, y) = \phi(x) - \phi(y) - \langle x - y, \nabla\phi(y) \rangle.$$

See Appendix A for common choices of Bregman divergence, their corresponding $\phi$, and $\text{dom}(\phi)$, which are

frequently used in clustering. For more examples see (Banerjee et al., 2005, Table 1), (Bauschke et al., 2016, Example 1), and (Ackermann, 2009, Figure 2.2). These examples underscore the versatility of Bregman divergences in accommodating various data characteristics and application domains. In Definition 2.1, the Bregman divergence was written as $D_\phi$, highlighting the underlying function $\phi$ it is generated from. For the sake of simplicity, we will continue to denote dissimilarity measures as $D$, with its underlying function $\phi$ being assumed.

The next proposition follows directly from (Banerjee et al., 2005, Proposition 1).

**Proposition 2.2.** *For $X \subset int\, dom(\phi)$, assume that for a $P \in S_2$, $s_k(P) > 0$ for a cluster $k \in [K]$. The unique optimal center for cluster $k$, $(c_P^*)_k$, for all $C_P^* \in C^*(P)$ equals*

$$(c_P^*)_k = \frac{\sum_{n=1}^{N} p_{k,n} w_n x_n}{\sum_{n=1}^{N} p_{k,n} w_n}. \qquad (4)$$

The proposition assumes that $X \subset \text{int dom}(\phi)$ to ensure that $(c_P^*)_k \in \text{int dom}(\phi)$, such as when clusters only contain a single point. We also note that $C^*(P)$ is a singleton unless there exists an empty cluster $k$ (i.e., $s_k(P) = 0$), for which any value of $(c_P^*)_k \in R$ is optimal.

From Definition 2.1, it must hold that $R \subseteq \text{int dom}(\phi)$. Following Proposition 2.2, we will also assume that $X \subset \text{int dom}(\phi)$. We note that this has no effect for Bregman divergences where $\text{dom}(\phi)$ is an open set, such as for squared Euclidean distance, squared Mahalanobis distance, and the Itakura-Saito divergence (see Appendix A). We can then further restrict $R = \text{co}(X)$ without any loss in solution quality given (4).

### 2.3. Local Optimality Conditions

Before defining local optimality, for $P \in S_1$, let $T(P)$ denote the set of adjacent points of $P$. The number of elements in $T(P)$ is finite; more specifically, for each $P \in S_1$, $|T(P)| = N(K - 1)$. This can be seen by considering all $S_1$-feasible single-step reassignments. Since each of the $N$ assigned points can be moved to one of the remaining $K - 1$ clusters, there are $N(K - 1)$ adjacent points.

We now introduce two definitions of local optimality for discrete and continuous optimization problems (Newby & Ali, 2015, Definitions 2 and 3) for problems (P1) and (P2), respectively, with respect to the function $F$.

**Definition 2.3** (D-local: Discrete locally optimal solution)**.** In (P1), $(P^*, C_{P^*}^*)$ is called D-local if the cluster assignment $P^*$ satisfies

$$F(P^*) \leq F(P) \quad \forall P \in T(P^*). \qquad (5)$$

**Definition 2.4** (C-local: Continuous locally optimal solution). In (P2), $(P^*, C^*_{P^*})$ is called C-local if there exists an $\epsilon > 0$ such that

$$F(P^*) \leq F(P) \quad \forall P \in S_2 \cap B(P^*, \epsilon). \tag{6}$$

We observe that D-local is a stronger notion of local optimality.

**Proposition 2.5** (Benson, 1995, p.82 without proof). *If $P^*$ is D-local, then it is C-local.*

Since we could not find a proof of the above proposition, we have included it in Appendix B, with the proofs of the remaining claims of this section, including the following extension of (Peng & Xia, 2005, Corollary 2.1).

**Proposition 2.6.** *For D-local and C-local solutions, no empty clusters exist.*

Our definitions of locally optimal solutions with respect to $F$ are motivated by the fact that given any $P \in S_2$, the optimal center $(c^*_P)_k$ for a non-empty cluster $k$ is given in closed form (4), making it natural to view (P1) and (P2) as optimization problems with respect to only the decision variables $P$. We can also consider locally optimal solutions of (P2) with respect to both $P$ and $C$ using the following definition.

**Definition 2.7** (CJ-local: Continuously and Jointly locally optimal solution). In (P2), $(P^*, C^*)$ is called CJ-local if there exists an $\epsilon > 0$ such that

$$f(P^*, C^*) \leq f(P, C) \quad \forall P \in S_2 \cap B(P^*, \epsilon) \quad \text{and}$$
$$\forall C \in R^K \cap B(C^*, \epsilon).$$

In general, C-local is a stronger definition of local optimality than CJ-local.

**Proposition 2.8.** *If $(P^*, C^*_{P^*})$ is C-local, then it is CJ-local.*

If a K-means algorithm (Section 3) converges, it will be to what is known as a partial optimal solution.

**Definition 2.9** (Partial optimal solution, Wendell & Hurter Jr, 1976, Equation (3)). In (P2), we define $(P^*, C^*_{P^*})$ as a partial optimal solution if it satisfies the following condition.

$$f(P^*, C^*_{P^*}) \leq f(P, C^*_{P^*}) \quad \forall P \in S_2. \tag{7}$$

## 3. K-means Algorithm

The K-means algorithm is a fast and practical heuristic algorithm which operates as follows (Lloyd, 1982).

Step 1 Select $K$ initial centers arbitrarily from $X$.

Step 2 For each $n \in [N]$, assign $x_n$ to a cluster $k \in [K]$ that minimizes $D(x_n, c_k)$: $k \in \arg\min_{k' \in [K]} D(x_n, c_{k'})$.

Step 3 For each non-empty $k \in [K]$, update the cluster center $c_k$ following (4).

Step 4 Repeat Step 2 and Step 3 until the cluster assignments $P$ no longer change.

A perhaps overlooked detail in Step 2 is that $\arg\min_{k' \in [K]} D(x_n, c_k)$ is in general not a singleton. Following (NumPy Developers, 2025, Notes), in this work the minimum index is always selected (see Algorithm 1, Line 5), with our analysis easily extending to any other deterministic selection rule.

### 3.1. Convergence to a Locally Optimal Solution Counterexample

As a counterexample to the K-means algorithm always converging to a locally optimal solution, we consider a 1-dimensional problem where all of the weights are equal to 1. We note that the optimal solution for the K-means problem can be solved in polynomial time and space using dynamic programming when the dimension of the data equals 1 (Wang & Song, 2011; Grønlund et al., 2017; Dupin & Nielsen, 2023).

Consider the situation where the number of points $N = 5$, the number of clusters $K = 2$, the dissimilarity measure $D(x, y) = \|x - y\|^2_2$, and the given dataset and initial centers are as follows.

$$x_1 = -4, \; x_2 = -2, \; x_3 = 0, \; x_4 = 1.5, \; x_5 = 2.5,$$

$$c_1 = x_3 = 0, \; c_2 = x_5 = 2.5.$$

Following Step 1-Step 4, the cluster assignments and centers converge to

$$P^* = \begin{bmatrix} 1 & 1 & 1 & 0 & 0 \\ 0 & 0 & 0 & 1 & 1 \end{bmatrix}, \quad c^*_1 = -2, \; c^*_2 = 2,$$

where all of the steps can be found in Appendix C. However, in the continuously relaxed problem (P2), $p_{1,3}$ can be moved towards $p_{2,3}$, resulting in

$$\hat{P} = \begin{bmatrix} 1 & 1 & 1-\alpha & 0 & 0 \\ 0 & 0 & \alpha & 1 & 1 \end{bmatrix},$$

$$\hat{c}_1 = \frac{-6}{3 - \alpha}, \text{ and } \hat{c}_2 = \frac{4}{2 + \alpha} \quad (0 < \alpha \leq 1).$$

The clustering loss can be written out as

$$f(\hat{P}, \hat{C}) = \frac{4(5\alpha - 6)}{\alpha - 3} + \frac{8.5\alpha^2 + 18\alpha + 2}{(\alpha + 2)^2},$$

and differentiating $f(\hat{P}, \hat{C})$ with respect to $\alpha$,

$$\frac{\mathrm{d}}{\mathrm{d}\alpha} f(\hat{P}, \hat{C}) = \frac{-20\alpha(\alpha + 12)}{(\alpha - 3)^2(\alpha + 2)^2} < 0, \quad (0 < \alpha \le 1).$$

Since this is negative for all $0 < \alpha \le 1$, and $f(\hat{P}, \hat{C})$ is continuous with respect to $\alpha$, $f(\hat{P}, \hat{C})$ is strictly decreasing for all $0 < \alpha \le 1$. Further, since $\hat{P}$ and $\hat{C}$ are both continuous with respect to $\alpha$, for any $\epsilon > 0$ in Definition 2.7, choosing $\alpha = \min(\alpha_P, \alpha_C)$ for $\alpha_P, \alpha_C > 0$ such that $\hat{P}(\alpha_P) \in S_2 \cap B(P, \epsilon)$ and $\hat{C}(\alpha_C) \in R^K \cap B(C, \epsilon)$ shows that $(P, C)$ is not a CJ-local solution. From Propositions 2.8 and 2.5, it follows that $(P, C)$ is also not C-local, nor D-local (see Appendix C for the full details).

### 3.2. Previous Research on Local Optimality

Until now, the only rigorous attempt at trying to prove that the K-means algorithm converges to a CJ-local solution seems to be in (Selim & Ismail, 1984). However, a critical error exists in their Lemma 7. The issue arises because the local optimality condition described in Lemma 7 only holds in general when $F$ is convex, e.g. (Bazaraa et al., 2006, Theorem 3.4.3). However, as already mentioned, it was proven in their work that $F$ is a concave function. Consequently, even in the widely studied case where the dissimilarity measure is the squared Euclidean distance, it is possible to construct counterexamples, such as the one discussed above, where the K-means algorithm fails to converge to a locally optimal solution.

## 4. Locally-Optimal K-means Algorithm

In this section, we discuss the necessary and sufficient conditions for locally optimal solutions, and propose a modified K-means algorithm that converges to a locally optimal solution according to Definitions 2.3 & 2.4. The proofs of the claims in this section are all in Appendix D.

### 4.1. Necessary and Sufficient Conditions for Locally Optimal Solutions

The following technical lemma describes the effect on the clustering loss $F$ when a point is moved from one cluster to another.

**Lemma 4.1.** *When the cluster that point $g \in [N]$ is assigned to changes from $a \in [K]$ to $b \in [K]$ by an amount $\alpha$ ($0 \le \alpha \le 1$), the change in $F$ equals*

$$\Delta_\alpha(g, a, b) := \alpha w_g\big(D(x_g, (c_P^*)_b) - D(x_g, (c_P^*)_a)\big)$$
$$- \big((s_a(P) - \alpha w_g)D((c_{P^{new}}^*)_a, (c_P^*)_a)$$
$$+ (s_b(P) + \alpha w_g)D((c_{P^{new}}^*)_b, (c_P^*)_b)\big), \tag{8}$$

*where $P$ and $P^{new}$ represent the cluster assignments before and after the change, respectively.*

---

**Function 1** Guarantees Convergence to C-local

1: **function** C-LO($X, W, P, C, D$)
2:     **for** $n = 1, 2, \ldots, N$ **do**
3:         **if** $|\arg\min_{k' \in [K]} D(x_n, c_{k'})| > 1$ **then**
4:             $k_1 \leftarrow \min(\arg\min_{k' \in [K]} D(x_n, c_{k'}))$
5:             $k_2 \leftarrow \max(\arg\min_{k' \in [K]} D(x_n, c_{k'}))$
6:             $p_{k_1,n} \leftarrow 0, p_{k_2,n} \leftarrow 1$
7:             Recalculate $c_{k_1}, c_{k_2}$
8:     **Return**

---

**Function 2** Guarantees Convergence to D-local

1: **function** D-LO($X, W, P, C, D$)
2:     **for** $n = 1, 2, \ldots, N$ **do**
3:         $k_1 \leftarrow \min(\arg\min_{k' \in [K]} D(x_n, c_{k'}))$
4:         **for** $k_2 = 1, 2, \ldots, k_1 - 1, k_1 + 1, \ldots, K$ **do**
5:             **if** $\Delta_1(n, k_1, k_2) < 0$ **then**
6:                 $p_{k_2,n} \leftarrow 1$
7:                 Recalculate $c_{k_2}$
8:                 **if** $s_{k_1}(P) = w_n$ **then**
9:                     $p_{k_1,n} \leftarrow 0$
10:                 **else**
11:                   $p_{k_1,n} \leftarrow 0$
12:                   Recalculate $c_{k_1}$
13:     **Return**

---

**Theorem 4.2.** *Suppose $(P, C_P^*)$ is a partial optimal solution and all clusters are non-empty. If $A(C_P^*)$ in (3) is a singleton, then $(P, C_P^*)$ is a C-local solution.*

**Theorem 4.3.** *Suppose all cluster centers are distinct for a solution $(P, C_P^*)$. If $(P, C_P^*)$ is a C-local solution, then $A(C_P^*)$ is a singleton. If $A(C_P^*)$ is not a singleton, then $A(C_P^*) \cap T(P)$ is not empty, and transitioning the cluster assignment from $P$ to any element $P' \in A(C_P^*) \cap T(P)$ guarantees a strict decrease in the clustering loss, $F(P') < F(P)$.*

Theorem 4.2 states that if the K-means algorithm converges and all of its clusters are non-empty, then if $A(C_P^*)$ is unique, $(P, C_P^*)$ is a C-local solution. Furthermore, if the K-means algorithm were to converge to a solution such that $A(C_P^*)$ is not a singleton and all of its cluster centers are distinct, Theorem 4.3 shows that choosing any adjacent point in $A(C_P^*)$ results in a strictly lower clustering loss.

In contrast to the preceding theorems characterizing C-local solutions, the necessary and sufficient conditions for D-local solutions in our analysis are simply using Definition 2.3, which can be verified directly over a given solution's finite adjacent points.

**Algorithm 1** Locally-Optimal K-means Algorithm (LO-K-means)

---

**Input:** $X = \{x_n\}_{n \in [N]} \subset \text{int dom}(\phi) \subseteq \mathbb{R}^d$, $W = \{w_n\}_{n \in [N]} \subset \mathbb{R}_{++}$, number of clusters $K \in \mathbb{N}$

1: Sample without replacement $\{c_k\}_{k \in [K]} \subset \{x_n\}_{n \in [N]}$ **(Step 1)**
2: Initialize $p_{k,n} \leftarrow 0$ for all $k \in [K]$, $n \in [N]$
3: **while** $P$ continues to change **do**
4:     $p_{k,n} \leftarrow 0$ for all $k \in [K]$, $n \in [N]$
5:     $p_{k,n} \leftarrow 1$ for all $n \in [N]$,
          $k = \min(\arg\min_{k' \in [K]} D(x_n, c_{k'}))$ **(Step 2)**
6:     **if** an empty cluster $a \in [K]$ exists **then**
7:         Move a point $g \in [N]$ from a cluster $b \in [K]$ to $a$, such that $s_b(P) > w_g$ and $x_g \neq c_b$.
8:     $c_k \leftarrow \frac{\sum_{n=1}^N p_{k,n} w_n x_n}{\sum_{n=1}^N p_{k,n} w_n}$ for all $k \in [K]$    **(Step 3)**
9:     **if** $P$ has not changed **then**         **(New Step)**
10:         **Case 1:** Function 1 - C-LO $(X, W, P, C, D)$
                                    ▷ guarantees C-local
11:         **Case 2:** Function 2 - D-LO $(X, W, P, C, D)$
                                    ▷ guarantees D-local

**Output:** $P$: cluster assignment, $C$: cluster centers

---

### 4.2. Modification to the K-means Algorithm

Now we propose a modified, Locally-Optimal K-means algorithm (LO-K-means) in Algorithm 1. Motivated by the above results, either Function 1 or 2 is invoked when the K-means algorithm converges. In Case 1, if $|A(C_P^*)| > 1$, Function 1 strictly decreases the clustering loss and prevents the algorithm from converging to a non-C-local solution using Theorem 4.3.

Similarly, in Case 2, Function 2 prevents Algorithm 1 from converging to a solution which is not D-local. We first note that for any cluster assignment $P$, any $P' \in T(P)$ can be generated as follows. For a $p_{a,g} = 1$, and an $a \neq b$, set

$$\begin{cases} p'_{a,g} = 0, & p'_{b,g} = 1, \\ p'_{k,n} = p_{k,n} \; \forall (k,n) \notin \{(a,g),(b,g)\}. \end{cases}$$

In addition, the resulting change in the clustering loss, $\Delta_1(g, a, b)$, can be computed using Lemma 4.1. Function 2 therefore searches for a point in $T(P)$ which strictly decreases the clustering loss, while preventing the algorithm from converging to a non-D-local solution.

**Theorem 4.4.** *Algorithm 1 converges to a C-local or D-local solution in a finite number of iterations when Case 1 or Case 2 is chosen, respectively.*

### 4.3. Method Variants and Comparisons

Both Functions 1 and 2 exit after finding the first adjacent point which guarantees a decrease in the clustering loss. Our convergence analysis (Theorem 4.4) holds when any

$(n, k_2)$ is chosen which would result in a decrease in the clustering loss. This is perhaps most interesting for Function 2, as the exact decrease $\Delta_1(n, k_1, k_2)$ is computed. In our experiments in Section 5, we also test a different variant of Function 2, Min-D-LO, which finds an $(n, k_2)$ pair which minimizes $\Delta_1$, moving the cluster assignment to an adjacent vertex that minimizes the clustering loss. A detailed implementation of Min-D-LO can be found as Function 3 in Appendix E.1.

The difference between our methods and the K-means algorithm occurs only after the K-means algorithm has converged. If the K-means algorithm has converged to a C or D-local solution, this can now be easily verified by a single call to Function 1 or 2. If the K-means algorithm does not converge to a locally optimal solution, our methods will perform additional iterations which are all guaranteed to strictly decrease the clustering loss. In particular, for any fixed iteration budget, our methods will always perform as well or better than the K-means algorithm (see Figure 3).

Given that our method is a simple modification of the K-means algorithm, it is highly compatible and complementary with the vast number of methods that either use the K-means algorithm as a subroutine or improve it in their own way. This includes, for example, works trying to find the optimal choice for $K$ (Pelleg & Moore, 2000; Hamerly & Elkan, 2003), methods on how to initialize the $K$ cluster means (Kanungo et al., 2004; Arthur & Vassilvitskii, 2006), as well as techniques to accelerate the K-means algorithm, such as Elkan's algorithm (Elkan, 2003).

### 4.4. Computational Complexity Analysis

For the K-means algorithm using squared Euclidean distance, the per-iteration time complexity is $O(NKd)$. The computation of $D$ is dependent on the specific Bregman divergence. Let $O(\Gamma_\phi(d))$ denote the time complexity of computing $D$ based on its underlying function $\phi$. Considering the examples given in Appendix A, it can be shown that $\Gamma_\phi(d) = d$ for the squared Euclidean distance, KL divergence, and Itakura-Saito divergence, while for the squared Mahalanobis distance, $\Gamma_\phi(d) = d^2$. In our analysis we will assume that $d = O(\Gamma_\phi(d))$. For general Bregman divergences, the per-iteration time complexity of the K-means algorithm becomes $O(NK\Gamma_\phi(d))$.

The following theorem considers the time complexity of the new steps introduced in Algorithm 1, and verifies that Algorithm 1 maintains a per-iteration time complexity of $O(NK\Gamma_\phi(d))$ (empirical analysis of computation time can be found in Section 5). This is achieved by using the following implementation details to precompute $s_k(P) = \sum_{n=1}^N p_{k,n} w_n$ for $k \in [K]$, which is used to recalculate $c_{k_1}$ and $c_{k_2}$ in Functions 1 and 2.

In Algorithm 1 at Line 4, $s_k(P)$ is initialized as $s_k(P) \leftarrow 0$ for all $k \in [K]$, and at Line 5, for all $n \in N$ and $k = \min(\arg\min_{k' \in [K]} D(x_n, c_{k'}))$, $s_k(P)$ is updated as $s_k(P) \leftarrow s_k(P) + p_{k,n} w_n$.

**Theorem 4.5.** *The per-iteration time complexity of Algorithm 1 equals $O(NK\Gamma_\phi(d))$ when calling C-LO or (Min-)D-LO.*

The space complexity of the K-means algorithm is $O((N+K)d)$ when using the squared Euclidean distance. Algorithm 1 continues to use the cluster assignment matrix $P = O(NK)$ introduced in Section 2. Following, for example, the implementation in scikit-learn, $P$ can be replaced by an array of labels of size $N$ where $P_n \in [K]$ indicates which cluster point $n$ is assigned to, e.g. Line 6 in Function 1 would become $P_n = k_2$. Using a more efficient method of representing cluster assignments, and noting that the precomputed array $s(P) = O(K)$, Algorithm 1 can be implemented with $O((N+K)d)$ space complexity. Using a general Bregman divergence also does not increase the space complexity when excluding its parameters, which seems to only concern the squared Mahalanobis distance, with its matrix $A \in \mathbb{R}^{d \times d}$, with all other examples of Bregman divergences within the references given in Section 2.2 requiring at most a single scalar parameter.

# 5. Experiments

We conducted experiments using both synthetic and real-world datasets to validate the performance of the LO-K-means algorithm (Algorithm 1). The experiments primarily used the squared Euclidean distance as the dissimilarity measure $D$, while experiments with other Bregman divergences (KL divergence & Itakura-Saito divergence), are presented in Appendix F.3.

## 5.1. Algorithms

We compared the following algorithms.

- K-means: LO-K-means algorithm (Algorithm 1) without the new step (Lines 9–11) (see Algorithm 2 in Appendix E.1).

- C-LO: Case 1 of the LO-K-means algorithm calling Function 1.

- D-LO: Case 2 of the LO-K-means algorithm calling Function 2.

- Min-D-LO: Case 2 of the LO-K-means algorithm calling a different variant of Function 2 (see Function 3 in Appendix E.1).

For the initialization of cluster centers in the above algorithms, we use two methods: uniform random sampling of $K$ points without replacement and the K-means++

method (Arthur & Vassilvitskii, 2006), as implemented in scikit-learn. When using K-means++ as the initialization method, we call the above algorithms K-means++, C-LO++, D-LO++, and Min-D-LO++, respectively, while for algorithms with the uniform random sampling, we use the above algorithm names. A comparison with the D-local algorithm proposed by Peng & Xia (2005) is presented in Appendix F.4.

For both the synthetic and real-world datasets, controlled experiments were conducted when comparing algorithms, guaranteeing that the dataset and initial cluster centers remained identical across all algorithms in each trial.

## 5.2. Synthetic Datasets

The synthetic datasets were generated as follows. We uniformly sampled $N$ data points from the space $[1, 10]^d$, restricted to integer values. If the same point is selected multiple times, the number of selections is assigned as its weight.

The experimental results comparing K-means with C-LO, and K-means++ with C-LO++ are shown in Figures 1 and 2 (see the additional experiments in Appendix F.1). Each figure presents the results when varying the number of data points $N$ and the number of clusters $K$, with 1,000 runs conducted for each experiment.

As shown in Figure 1, in many cases, particularly in low-dimensional settings, K-means does not converge to a C-local solution, regardless of the initialization. Additionally, when the number of clusters $K$ is large relative to the number of data points $N$, or when data points are densely distributed in the space, K-means often fails to converge to a C-local solution. In contrast, the C-LO algorithm consistently achieves a C-local solution with lower clustering loss.

Figure 2 illustrates the improvement ratio of the clustering loss, given by $(F(P) - F(P_{LO}))/F(P)$, where $P_{LO}$ is the output of C-LO and $P$ is the output of K-means. It can be observed that the improvement tends to be more significant in lower-dimensional settings.

## 5.3. Real-World Datasets

### 5.3.1. METRIC

We also did experiments using real-world datasets (see the details of each dataset used in Appendix E.2). Given the randomness in the initial cluster center selection, each algorithm was executed 20 times. The performance metrics used for comparison include the average, variance, and minimum values of the clustering loss, as well as the average computation time and the average number of iterations of the algorithm. The evaluation was conducted for cluster

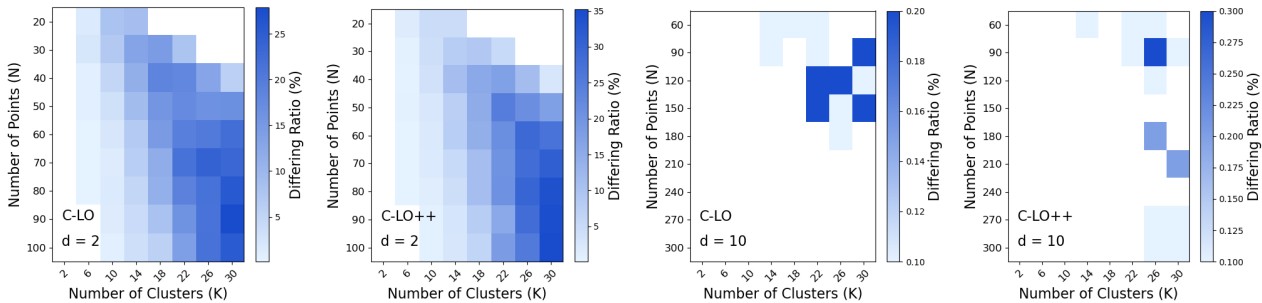

*Figure 1.* The proportion of cases where the clustering loss is improved over K-means by using C-LO across two different initialization methods and dimensions, with $D$ equal to the squared Euclidean distance. These plots indicate the proportion of instances where K-means did not converge to a C-local solution. Each $N, K$ cell represents the results from 1,000 runs of both algorithms. Darker colors indicate a higher frequency of clustering loss improvement.

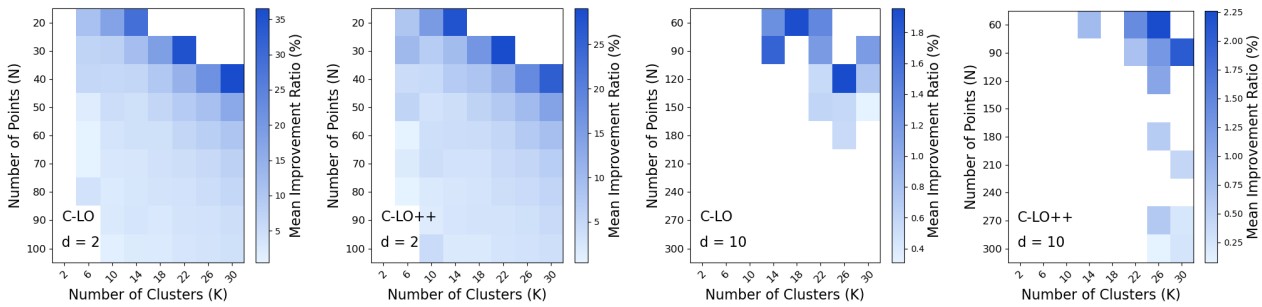

*Figure 2.* The average improvement ratio of the clustering loss when C-LO improves over K-means across two different initialization methods and dimensions, with $D$ equal to the squared Euclidean distance. Each $N, K$ cell represents the results from 1,000 runs of both algorithms. Darker colors indicate a higher percentage of clustering loss improvement.

sizes of $K = 5, 10, 25,$ and $50$.

### 5.3.2. RESULTS

In real-world datasets, where features are real numbers, it is rare for $A(C)$ in (3) to consist of multiple elements. Consequently, C-LO did not yield improvements in the clustering loss (details in Tables 2–11 in Appendices F.2–F.4). These results suggest that although the K-means algorithm was not explicitly designed to guarantee this property, it seems to be able to converge to a C-local solution in most real-world datasets. However, it is observed that the time to verify that the K-means algorithm has converged to a C-local solution using C-LO is negligible, with no measured increase in computation time using C-LO compared to K-means in the vast majority of experiments.

On the other hand, as shown in Table 1, D-LO and Min-D-LO consistently achieve a lower clustering loss compared to K-means. We also observe that in general, achieving a D-local optimal solution requires more computation time. Although the per-iteration time complexity remains unchanged for all algorithms, D-LO and Min-D-LO require more iterations to converge. In order to mitigate this problem, we first observe that D-LO and Min-D-LO are

highly compatible with initializing cluster centers using K-means++, reducing their number of iterations in the majority of instances similarly to K-means.

The slower convergence time of D-LO compared to K-means is also what motivated the initial search for other variants of Function 2, which resulted in Min-D-LO. As both D-LO and Min-D-LO converge to D-local optimal solutions, we observe that their accuracy is similar, while the number of iterations Min-D-LO requires is never greater than, and at times much smaller than required of D-LO, resulting in a significantly reduced runtime.

A common way to balance accuracy and computation time is to enforce an iteration limit. We note that in all of our experiments, we ran the algorithms until convergence, which may have resulted in excessive time spent for only marginally better solutions. Figure 3 illustrates the clustering loss after each iteration of D-LO and Min-D-LO, including their common K-means iterations before Function 2 or 3 was first called. We observe that limiting the number of iterations to 200–300 leads to substantial improvements in computational efficiency, with a speedup of up to $5\times$ for D-LO and $2\times$ for Min-D-LO, while still always ensuring an accuracy improvement of over 15% for D-LO and over

*Table 1.* Mean, variance, and minimum of the clustering loss, along with the average computation time and average number of iterations over 20 runs for each initialization method and number of clusters, for K-means, D-LO, and Min-D-LO, with $D$ chosen as the squared Euclidean distance, for three real-world datasets.

| | Dataset | Iris ($N=150, d=4$) | | | | Wine Quality ($N=6,497, d=11$) | | | | News20 - 1 ($N=2,000, d=1,089$) | | | |
|---|---|---|---|---|---|---|---|---|---|---|---|---|---|
| $K$ | Algorithm | Mean ± Variance | Minimum | Time(s) | Num Iter | Mean ± Variance | Minimum | Time(s) | Num Iter | Mean ± Variance | Minimum | Time(s) | Num Iter |
| 5 | K-means | 57.54 ± 9.78 | **46.54** | < 0.001 | 9 | 2,393,869 ± 67 | 2,393,751 | 0.005 | 40 | 984,655 ± 55,781 | 864,073 | 0.31 | 31 |
| | D-LO | **57.32** ± 9.83 | **46.54** | < 0.001 | 13 | **2,393,759** ± 12 | **2,393,742** | 0.006 | 49 | 808,391 ± 0 | 808,391 | 1.94 | 168 |
| | Min-D-LO | **57.32** ± 9.83 | **46.54** | < 0.001 | 13 | **2,393,759** ± 12 | **2,393,742** | 0.006 | 49 | **808,032** ± 1,566 | **801,207** | 0.97 | 81 |
| | K-means++ | 50.58 ± 4.74 | **46.54** | < 0.001 | 8 | 2,499,611 ± 211,445 | 2,393,754 | 0.005 | 40 | 870,899 ± 80,013 | 808,450 | 0.20 | 18 |
| | D-LO++ | **50.30** ± 4.70 | **46.54** | < 0.001 | 13 | **2,393,753** ± 11 | **2,393,742** | 0.008 | 63 | **806,236** ± 3,292 | **801,207** | 0.82 | 74 |
| | Min-D-LO++ | **50.30** ± 4.70 | **46.54** | < 0.001 | 13 | **2,393,753** ± 11 | 2,393,746 | 0.008 | 62 | 806,595 ± 3,111 | **801,207** | 0.53 | 48 |
| 10 | K-means | 31.55 ± 5.42 | 26.78 | < 0.001 | 8 | 1,378,087 ± 6,569 | 1,367,222 | 0.01 | 51 | 919,058 ± 73,594 | 734,571 | 0.71 | 35 |
| | D-LO | **30.53** ± 5.00 | **26.18** | < 0.001 | 20 | **1,377,711** ± 6,571 | **1,367,222** | 0.02 | 74 | **637,004** ± 4,319 | **625,281** | 19.52 | 870 |
| | Min-D-LO | 30.55 ± 4.99 | **26.18** | < 0.001 | 19 | **1,377,711** ± 6,571 | **1,367,222** | 0.02 | 74 | 642,980 ± 7,605 | 637,400 | 6.65 | 317 |
| | K-means++ | 29.57 ± 2.97 | 26.01 | < 0.001 | 7 | 1,381,737 ± 9,516 | 1,365,020 | 0.01 | 52 | 697,527 ± 32,211 | 643,583 | 0.48 | 23 |
| | D-LO++ | **28.92** ± 3.00 | **25.94** | < 0.001 | 17 | **1,380,941** ± 9,674 | **1,364,944** | 0.02 | 87 | **634,216** ± 5,596 | **625,467** | 6.18 | 288 |
| | Min-D-LO++ | 28.93 ± 3.00 | **25.94** | < 0.001 | 17 | **1,380,941** ± 9,674 | **1,364,944** | 0.02 | 84 | 634,293 ± 6,477 | 625,468 | 2.55 | 125 |
| 25 | K-means | 15.98 ± 1.64 | 13.67 | < 0.001 | 7 | 681,397 ± 24,123 | 659,902 | 0.04 | 86 | 790,822 ± 88,437 | 650,038 | 1.65 | 34 |
| | D-LO | 14.59 ± 1.68 | **11.89** | < 0.001 | 36 | **665,882** ± 7,059 | **654,126** | 0.12 | 235 | **481,983** ± 5,198 | 475,651 | 155.39 | 3,016 |
| | Min-D-LO | **14.49** ± 1.76 | 12.02 | < 0.001 | 30 | 666,641 ± 7,082 | **654,126** | 0.10 | 202 | 485,809 ± 6,874 | **473,159** | 40.17 | 787 |
| | K-means++ | 13.73 ± 0.68 | 12.70 | < 0.001 | 6 | 654,219 ± 15,173 | 631,244 | 0.03 | 52 | 529,028 ± 32,301 | 487,823 | 1.25 | 26 |
| | D-LO++ | 12.58 ± 0.45 | **11.83** | < 0.001 | 31 | 649,046 ± 14,736 | 630,546 | 0.06 | 121 | 475,299 ± 3,831 | 468,201 | 35.96 | 705 |
| | Min-D-LO++ | **12.61** ± 0.43 | 12.07 | < 0.001 | 27 | **649,001** ± 14,774 | 631,157 | 0.05 | 107 | **474,431** ± 4,508 | **467,745** | 15.77 | 316 |
| 50 | K-means | 8.65 ± 1.27 | 7.05 | < 0.001 | 5 | 434,056 ± 14,094 | 402,580 | 0.11 | 63 | 731,980 ± 91,535 | 552,717 | 2.72 | 28 |
| | D-LO | 7.13 ± 1.50 | 5.66 | 0.002 | 48 | **424,122** ± 20,475 | 381,856 | 0.28 | 259 | **400,826** ± 2,920 | **395,833** | 314.36 | 3,133 |
| | Min-D-LO | **7.09** ± 1.35 | **5.62** | 0.002 | 39 | 424,435 ± 20,610 | 382,197 | 0.29 | 229 | 402,716 ± 2,469 | 398,453 | 103.30 | 1,040 |
| | K-means++ | 6.40 ± 0.34 | 5.52 | < 0.001 | 5 | 376,544 ± 8,469 | 367,108 | 0.05 | 45 | 439,029 ± 10,015 | 418,754 | 3.02 | 31 |
| | D-LO++ | **5.36** ± 0.24 | **5.04** | 0.002 | 37 | **372,716** ± 5,701 | 365,447 | 0.21 | 188 | **392,016** ± 1,513 | **388,746** | 157.97 | 1,228 |
| | Min-D-LO++ | 5.40 ± 0.23 | **5.04** | 0.002 | 30 | 373,075 ± 5,877 | **364,596** | 0.18 | 164 | 392,146 ± 2,080 | 388,990 | 60.41 | 533 |

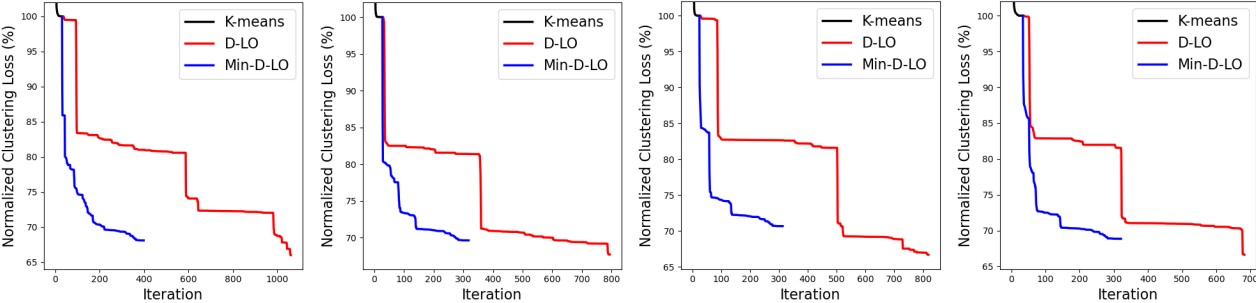

*Figure 3.* The clustering loss progression for four runs of D-LO, Min-D-LO, and their common K-means iterations on the News20 dataset with ($N = 2,000, d = 1,089$) with $K = 10$. The clustering losses are normalized such that the clustering loss achieved by the K-means algorithm upon convergence is set to 100%.

25% for Min-D-LO compared to K-means. We also note that this analysis is in line with the default iteration limit of 300 for the K-means algorithm used in scikit-learn.

Our final experiments compared our methods with the D-local algorithm proposed in (Peng & Xia, 2005, Section 3.2.1) (D-LO-P&X) in Appendix F.4. We observe that all D-local algorithms achieved similar values for the clustering loss which were significantly lower than the clustering loss of K-means, while D-LO-P&X was significantly slower than all of the tested methods, with Min-D-LO being in general much faster than the other tested D-local algorithms.

## 6. Conclusion

This work focused on the local optimality properties of the K-means algorithm. Even though it seems to be widely understood that the K-means algorithm converges to a locally optimal solution, we showed by simple counterexample that this is in fact not true. Motivated by this finding, we analyzed two definitions of local optimality, discrete (D-local) and continuous (C-local), suited for the K-means problem and its continuous relaxation. Considering a generalized weighted K-means problem using Bregman divergences, and guided by our theoretical analysis, a modified K-means algorithm was developed, LO-K-means, consisting of simple improvements to the K-means algorithm, guaranteeing it to converge to either a C-local or D-local solution. It was shown that LO-K-means is an efficient algorithm, matching the K-means algorithm's computational complexity. Improved empirical performance in terms of solution quality was also observed, in particular for our Min-D-LO algorithm variant, which consistently found solutions with a lower clustering loss on both synthetic and real-world datasets compared to the K-means algorithm, while also being fast relative to other D-local algorithms.

## Impact Statement

This paper presents work whose goal is to advance the field of Machine Learning. There are many potential societal consequences of our work, none which we feel must be specifically highlighted here.

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

## A. Examples of Bregman Divergences

We list common choices of Bregman divergence and their corresponding $\phi$, frequently used in clustering. For more examples see (Banerjee et al., 2005, Table 1), (Bauschke et al., 2016, Example 1), and (Ackermann, 2009, Figure 2.2).

- $\phi(x) = \|x\|_2^2$ with $\mathrm{dom}(\phi) = \mathbb{R}^d$. This represents the squared Euclidean distance, a fundamental and commonly applied dissimilarity measure ideal for spherical clustering applications (Steinbach et al., 2000; Berthold & Höppner, 2016; Capó et al., 2017).

- $\phi(x) = x^\top A x$ with $\mathrm{dom}(\phi) = \mathbb{R}^d$. This function corresponds to the squared Mahalanobis distance, which extends the squared Euclidean distance by incorporating variable correlations. It is widely used in big data and data mining domains (Brown et al., 2022; Martino et al., 2019).

- $\phi(x) = \sum_{i=1}^d x_i \log x_i$ with $\mathrm{dom}(\phi) = \mathbb{R}_+^d$, where $0 \log 0 = 0$ by convention. This is the Kullback-Leibler (KL) divergence (Kullback & Leibler, 1951), frequently applied in information theory and probabilistic clustering, particularly for image and text data (Wang et al., 2022; Dhillon et al., 2003).

- $\phi(x) = -\sum_{i=1}^d \log x_i$ with $\mathrm{dom}(\phi) = \mathbb{R}_{++}^d$. Known as the Itakura-Saito divergence (Itakura & Saito, 1968), this measure is especially valued in signal processing and audio analysis for its scale-invariant properties. It finds applications in areas such as speech coding (Linde et al., 1980; Battenberg et al., 2012).

These examples underscore the versatility of Bregman divergences in accommodating various data characteristics and application domains.

## B. Proofs Concerning Some Local Optimality Relations

***Proof of Proposition 2.5.*** There are $E_T := N(K-1)$ extreme points in $T(P^*)$ as discussed in Section 2.3, denoted as $P_i$ $(1 \le i \le E_T)$. Let $d_i := P_i - P^*$ for all $P_i \in T(P^*)$, which is a feasible direction in $S_2$. Since $P^*$ is D-local, according to (5), $F(P^*) \le F(P_i)$ holds.

Since $F(P)$ is a concave function, by Jensen's inequality, for any $z_1, \ldots, z_{E_T} \in S_2$,

$$a_1 F(z_1) + \cdots + a_{E_T} F(z_{E_T}) \le F(a_1 z_1 + \cdots + a_{E_T} z_{E_T}) \quad (a_1 + \cdots + a_{E_T} = 1,\ a_1, \ldots, a_{E_T} \ge 0). \tag{9}$$

From the fact that $F(P^*) \le F(P_i)$ and $F$ is concave, for any $0 \le \beta \le 1$, it holds that

$$F(P^*) \le (1-\beta)F(P^*) + \beta F(P^* + d_i) \le F((1-\beta)P^* + \beta(P^* + d_i)) = F(P^* + \beta d_i).$$

Now, we show that $F(P^*) \le F(P)$ for all $P \in S_2 \cap B(P^*, \epsilon)$. Any $P$ can be expressed as $P = P^* + \sum_{i=1}^{E_T} \beta_i d_i$ with some $\beta_i \ge 0$.

Letting $\gamma = \sum_{i=1}^{E_T} \beta_i$ (where $\gamma \le 1$ holds for sufficiently small $\epsilon$), substituting $z_i = P^* + \gamma d_i$ and $a_i = \frac{\beta_i}{\gamma}$ into (9), we obtain

$$F(P^*) \le \frac{\beta_1}{\gamma} F(P^* + \gamma d_1) + \cdots + \frac{\beta_{E_T}}{\gamma} F(P^* + \gamma d_{E_T}) \le F(P),$$

which satisfies (6), proving that $P^*$ is C-local.

Thus, $P^*$ being D-local is a sufficient condition for it to also be C-local. ∎

The following proposition is a general result which will be used to prove Proposition 2.6 as well as Lemma D.4.

**Proposition B.1.** *Assume a cluster assignment $P$, with an empty cluster $a$, and cluster centers $C$ are given, where $(P, C)$ is feasible in (P1). There exists a cluster $b$ containing at least two points, with one of the points, $x_g \ne c_b$. If the cluster assignment of $x_g$ is shifted from cluster $b$ to $a$ by a factor of $0 < \alpha \le 1$, and denoting the new cluster assignment as $P^{new}$, it holds that $F(P^{new}) < f(P, C)$.*

*Proof.* Since there are at least $K + 1$ unique points to be assigned to $K$ clusters, there exists a cluster $b$ containing at least two distinct points $x_g$ and $x_h$, such that $x_g \neq c_b$. After shifting $x_g$ from cluster $b$ to $a$ by a factor of $0 < \alpha \leq 1$, the optimal center of cluster $a$ becomes $(c^*_{P^{\text{new}}})_a = x_g$. Writing out $f(P, C)$ and $F(P^{\text{new}})$,

$$
\begin{aligned}
f(P, C) &= \sum_{k \notin \{a,b\}} \sum_{n=1}^{N} p_{k,n} w_n D(x_n, c_k) + \sum_{k \in \{a,b\}} \sum_{n=1}^{N} p_{k,n} w_n D(x_n, c_k) \\
&= \sum_{k \notin \{a,b\}} \sum_{n=1}^{N} p_{k,n}^{\text{new}} w_n D(x_n, c_k) + \sum_{n=1}^{N} p_{b,n} w_n D(x_n, c_b) \\
&= \sum_{k \notin \{a,b\}} \sum_{n=1}^{N} p_{k,n}^{\text{new}} w_n D(x_n, c_k) + \sum_{n \neq g} p_{b,n}^{\text{new}} w_n D(x_n, c_b) + ((1 - \alpha) + \alpha) w_g D(x_g, c_b) \\
&= \sum_{k \notin \{a,b\}} \sum_{n=1}^{N} p_{k,n}^{\text{new}} w_n D(x_n, c_k) + \sum_{n=1}^{N} p_{b,n}^{\text{new}} w_n D(x_n, c_b) + \alpha w_g D(x_g, c_b) \\
&= \sum_{k \neq a} \sum_{n=1}^{N} p_{k,n}^{\text{new}} w_n D(x_n, c_k) + \alpha w_g D(x_g, c_b), \text{ and} \\
F(P^{\text{new}}) &= \sum_{k=1}^{K} \sum_{n=1}^{N} p_{k,n}^{\text{new}} w_n D(x_n, (c^*_{P^{\text{new}}})_k) = \sum_{k \neq a} \sum_{n=1}^{N} p_{k,n}^{\text{new}} w_n D(x_n, (c^*_{P^{\text{new}}})_k).
\end{aligned}
$$

The change in the clustering loss is then equal to

$$
F(P^{\text{new}}) - f(P, C) = \sum_{k \neq a} \sum_{n=1}^{N} p_{k,n}^{\text{new}} w_n (D(x_n, (c^*_{P^{\text{new}}})_k) - D(x_n, c_k)) - \alpha w_g D(x_g, c_b).
$$

Since $C^*_{P^{\text{new}}}$ minimizes the clustering loss for the cluster assignment $P^{\text{new}}$,

$$
\sum_{k \neq a} \sum_{n=1}^{N} p_{k,n}^{\text{new}} w_n (D(x_n, (c^*_{P^{\text{new}}})_k) - D(x_n, c_k)) \leq 0.
$$

Moreover, since $x_g \neq c_b$, it follows that $D(x_g, c_b) > 0$ from Lemma D.1. Thus, $F(P^{\text{new}}) - f(P, C) < 0$ for any $\alpha > 0$. ∎

**Proof of Proposition 2.6.** Considering $(P, C)$ in Proposition B.1, with $C = C^*_P$, the clustering loss can be strictly decreased for any $\alpha > 0$, indicating that $P$ is not C-local. If we set $\alpha = 1$, it further shows that $P$ is not D-local either. ∎

**Proof of Proposition 2.8.** By (6), there exists an $\epsilon > 0$ such that

$$
F(P^*) = f(P^*, C^*_{P^*}) \leq F(P), \quad \forall P \in S_2 \cap B(P^*, \epsilon).
$$

From (2), we know that $F(P) \leq f(P, C)$ for all $C \in R^K$. Therefore, it follows that

$$
f(P^*, C^*_{P^*}) \leq f(P, C), \quad \forall P \in S_2 \cap B(P^*, \epsilon) \text{ and } \forall C \in R^K.
$$

∎

# C. Detailed Counterexample of the K-means Algorithm Converging to a Locally Optimal Solution

Consider the situation where the number of points $N = 5$, the number of clusters $K = 2$, the dissimilarity measure $D(x,y) = \|x - y\|_2^2$, and the given dataset and initial centers are as follows.

$$x_1 = -4, \ x_2 = -2, \ x_3 = 0, \ x_4 = 1.5, \ x_5 = 2.5,$$

$$c_1 = x_3 = 0, \ c_2 = x_5 = 2.5.$$

**Iteration 1** By calculating the dissimilarity measures $d_{k,n} = D(x_n, c_k)$ for all $n \in [N]$ and $k \in [K]$,

$$d_{1,1} = 16, \quad d_{1,2} = 4, \quad d_{1,3} = 0, \quad d_{1,4} = 2.25, \quad d_{1,5} = 6.25,$$
$$d_{2,1} = 42.25, \quad d_{2,2} = 20.25, \quad d_{2,3} = 6.25, \quad d_{2,4} = 1, \quad d_{2,5} = 0.$$

Based on $\{d_{k,n}\}$, points are assigned to the nearest center,

$$P = \begin{bmatrix} 1 & 1 & 1 & 0 & 0 \\ 0 & 0 & 0 & 1 & 1 \end{bmatrix}.$$

Based on $P$, the clusters are recomputed as

$$c_1 = -2, \ c_2 = 2.$$

**Iteration 2** Repeating the same steps again,

$$d_{1,1} = 4, \quad d_{1,2} = 0, \quad d_{1,3} = 4, \quad d_{1,4} = 12.25, \quad d_{1,5} = 20.25,$$
$$d_{2,1} = 36, \quad d_{2,2} = 16, \quad d_{2,3} = 4, \quad d_{2,4} = 0.25, \quad d_{2,5} = 0.25.$$

Based on $\{d_{k,n}\}$, points are assigned to the nearest center,

$$P = \begin{bmatrix} 1 & 1 & 1 & 0 & 0 \\ 0 & 0 & 0 & 1 & 1 \end{bmatrix}.$$

Based on $P$, the clusters are recomputed as

$$c_1 = -2, \ c_2 = 2,$$

and the K-means algorithm has converged. However, in the continuously relaxed problem (P2), $p_{1,3}$ can be moved towards $p_{2,3}$, resulting in

$$\hat{P} = \begin{bmatrix} 1 & 1 & 1-\alpha & 0 & 0 \\ 0 & 0 & \alpha & 1 & 1 \end{bmatrix},$$

$$\hat{c}_1 = \frac{-6}{3-\alpha}, \text{ and } \hat{c}_2 = \frac{4}{2+\alpha} \quad (0 < \alpha \leq 1).$$

The clustering loss can be written out as

$$f(\hat{P}, \hat{C}) = \left( -4 - \frac{-6}{3-\alpha} \right)^2 + \left( -2 - \frac{-6}{3-\alpha} \right)^2 + (1-\alpha) \left( \frac{-6}{3-\alpha} \right)^2$$
$$+ \alpha \left( \frac{4}{2+\alpha} \right)^2 + \left( 1.5 - \frac{4}{2+\alpha} \right)^2 + \left( 2.5 - \frac{4}{2+\alpha} \right)^2$$
$$= \frac{4(5\alpha - 6)}{\alpha - 3} + \frac{8.5\alpha^2 + 18\alpha + 2}{(\alpha + 2)^2}.$$

Differentiating $f(\hat{P}, \hat{C})$ with respect to $\alpha$,

$$\frac{\mathrm{d}}{\mathrm{d}\alpha} f(\hat{P}, \hat{C}) = \frac{-20\alpha(\alpha + 12)}{(\alpha - 3)^2(\alpha + 2)^2} < 0 \quad (0 < \alpha \leq 1).$$

Since this is negative for all $0 < \alpha \leq 1$, and $f(\hat{P}, \hat{C})$ is continuous with respect to $\alpha$, $f(\hat{P}, \hat{C})$ is strictly decreasing for all $0 < \alpha \leq 1$. Further, since $\hat{P}$ and $\hat{C}$ are both continuous with respect to $\alpha$, for any $\epsilon > 0$ in Definition 2.7, choosing $\alpha = \min(\alpha_P, \alpha_C)$ for $\alpha_P, \alpha_C > 0$ such that $\hat{P}(\alpha_P) \in S_2 \cap B(P, \epsilon)$ and $\hat{C}(\alpha_C) \in R^K \cap B(C, \epsilon)$ shows that $(P, C)$ is not a CJ-local solution. From Propositions 2.8 and 2.5, it follows that $(P, C)$ is also not C-local, nor D-local.

## D. Proofs in Section 4

### D.1. Supporting Lemmas

In our analysis the following lemmas will be required to prove our main results.

**Lemma D.1** (Bregman, 1967, Page 200). *For all $(x, c) \in dom(\phi) \times int\ dom(\phi)$ $D(x, c) \geq 0$, and $D(x, c) = 0$ if and only if $x = c$.*

**Lemma D.2.** *For any two non-empty cluster assignments for a cluster $k \in [K]$, $\{p_{k,n}\}$ and $\{p'_{k,n}\}$, it holds that*

$$(c^*_{P'})_k = (c^*_P)_k + \frac{\sum_{n=1}^N (p'_{k,n} - p_{k,n}) w_n (x_n - (c^*_P)_k)}{\sum_{n=1}^N p'_{k,n} w_n}.$$

*Proof.* From Proposition 2.2,

$$(c^*_P)_k = \frac{\sum_{n=1}^N p_{k,n} w_n x_n}{\sum_{n=1}^N p_{k,n} w_n} \text{ and}$$

$$\begin{aligned}
(c^*_{P'})_k &= \frac{\sum_{n=1}^N p'_{k,n} w_n x_n}{\sum_{n=1}^N p'_{k,n} w_n} \\
&= \frac{\sum_{n=1}^N p_{k,n} w_n x_n + \sum_{n=1}^N (p'_{k,n} - p_{k,n}) w_n x_n}{\sum_{n=1}^N p'_{k,n} w_n} \\
&= \frac{\sum_{n=1}^N p_{k,n} w_n}{\sum_{n=1}^N p'_{k,n} w_n} (c^*_P)_k + \frac{\sum_{n=1}^N (p'_{k,n} - p_{k,n}) w_n x_n}{\sum_{n=1}^N p'_{k,n} w_n} \\
&= \left( 1 + \frac{\sum_{n=1}^N (p_{k,n} - p'_{k,n}) w_n}{\sum_{n=1}^N p'_{k,n} w_n} \right) (c^*_P)_k + \frac{\sum_{n=1}^N (p'_{k,n} - p_{k,n}) w_n x_n}{\sum_{n=1}^N p'_{k,n} w_n} \\
&= (c^*_P)_k + \frac{\sum_{n=1}^N (p'_{k,n} - p_{k,n}) w_n (x_n - (c^*_P)_k)}{\sum_{n=1}^N p'_{k,n} w_n}.
\end{aligned}$$

∎

### D.2. Proof of Lemma 4.1

*Proof of Lemma 4.1.* The new cluster assignment $P^{new}_{k,n}$ is defined as

$$p^{new}_{k,n} = \begin{cases} p_{k,n} - \alpha & \text{if } (k, n) = (a, g), \\ p_{k,n} + \alpha & \text{if } (k, n) = (b, g), \quad (0 \leq \alpha \leq 1) \\ p_{k,n} & \text{otherwise.} \end{cases} \tag{10}$$

In this case, the contributions from points belonging to clusters other than $a$ and $b$ remain unchanged. Let the changes in the function values within clusters $a$ and $b$ be denoted by $\Delta_a$ and $\Delta_b$, respectively. Beginning with cluster $a$,

$$\Delta_a = \sum_{n=1}^N p^{new}_{a,n} w_n D(x_n, (c^*_{P^{new}})_a) - \sum_{n=1}^N p_{a,n} w_n D(x_n, (c^*_P)_a).$$

From (10), we know that $p_{a,g} = p^{new}_{a,g} + \alpha$. Substituting this, we have

$$\Delta_a = \sum_{n=1}^N p^{new}_{a,n} w_n D(x_n, (c^*_{P^{new}})_a) - \sum_{n=1}^N p^{new}_{a,n} w_n D(x_n, (c^*_P)_a) - \alpha w_g D(x_g, (c^*_P)_a).$$

From Definition 2.1, we expand the terms as

$$\Delta_a = \sum_{n=1}^{N} p_{a,n}^{\text{new}} w_n \big( \phi(x_n) - \phi(c_{P\text{new}}^*)_a) - \langle x_n - (c_{P\text{new}}^*)_a, \nabla\phi((c_{P\text{new}}^*)_a) \rangle \big)$$

$$- \sum_{n=1}^{N} p_{a,n}^{\text{new}} w_n \big( \phi(x_n) - \phi((c_P^*)_a) - \langle x_n - (c_P^*)_a, \nabla\phi((c_P^*)_a) \rangle \big) - \alpha w_g D(x_g, (c_P^*)_a)$$

$$= \sum_{n=1}^{N} p_{a,n}^{\text{new}} w_n \big( \phi((c_P^*)_a) - \phi((c_{P\text{new}}^*)_a) - \langle x_n - (c_{P\text{new}}^*)_a, \nabla\phi((c_{P\text{new}}^*)_a) \rangle$$

$$+ \langle x_n - (c_P^*)_a, \nabla\phi((c_P^*)_a) \rangle \big) - \alpha w_g D(x_g, (c_P^*)_a).$$

From (4), we know that $\sum_{n=1}^{N} p_{a,n}^{\text{new}} w_n x_n = \sum_{n=1}^{N} p_{a,n}^{\text{new}} w_n (c_{P\text{new}}^*)_a$. Applying this property two times,

$$\Delta_a = \sum_{n=1}^{N} p_{a,n}^{\text{new}} w_n \big( \phi((c_P^*)_a) - \phi((c_{P\text{new}}^*)_a) + \langle (c_{P\text{new}}^*)_a - (c_P^*)_a, \nabla\phi((c_P^*)_a) \rangle \big) - \alpha w_g D(x_g, (c_P^*)_a).$$

Finally, substituting $s_a(P) - \alpha w_g = \sum_{n=1}^{N} p_{a,n}^{\text{new}} w_n$, where $s_a(P) = \sum_{n=1}^{N} p_{a,n} w_n$, we obtain

$$\Delta_a = (s_a(P) - \alpha w_g)(-D((c_{P\text{new}}^*)_a, (c_P^*)_a)) - \alpha w_g D(x_g, (c_P^*)_a).$$

Similarly, for $\Delta_b$, we have

$$\Delta_b = \sum_{n=1}^{N} p_{b,n}^{\text{new}} w_n D(x_n, (c_{P\text{new}}^*)_b) - \sum_{n=1}^{N} p_{b,n} w_n D(x_n, (c_P^*)_b)$$

$$= \sum_{n=1}^{N} p_{b,n}^{\text{new}} w_n D(x_n, (c_{P\text{new}}^*)_b) - \sum_{n=1}^{N} p_{b,n}^{\text{new}} w_n D(x_n, (c_P^*)_b) + \alpha w_g D(x_g, (c_P^*)_b)$$

$$= (s_b(P) + \alpha w_g)(-D((c_{P\text{new}}^*)_b, (c_P^*)_b)) + \alpha w_g D(x_g, (c_P^*)_b).$$

Therefore, the change in the clustering loss equals

$$F(P^{\text{new}}) - F(P) = \Delta_a + \Delta_b$$

$$= (s_a(P) - \alpha w_g)(-D((c_{P\text{new}}^*)_a, (c_P^*)_a)) - \alpha w_g D(x_g, (c_P^*)_a)$$

$$+ (s_b(P) + \alpha w_g)(-D((c_{P\text{new}}^*)_b, (c_P^*)_b)) + \alpha w_g D(x_g, (c_P^*)_b)$$

$$= \alpha w_g (D(x_g, (c_P^*)_b) - D(x_g, (c_P^*)_a))$$

$$- \big( (s_a(P) - \alpha w_g) D((c_{P\text{new}}^*)_a, (c_P^*)_a) + (s_b(P) + \alpha w_g) D((c_{P\text{new}}^*)_b, (c_P^*)_b) \big).$$

∎

### D.3. Proofs of Theorem 4.2 & Theorem 4.3

The following Proposition D.3 is used in the proof of Theorem 4.2.

**Proposition D.3** (Wendell & Hurter Jr, 1976, Corollary 2)**.** *For problem (P2), suppose* $(P^*, C_{P^*}^*)$ *is a partial optimal solution. If* $A(C_{P^*}^*)$ *is a singleton, then* $(P^*, C_{P^*}^*)$ *is a CJ-local solution.*

*Proof.* We verify that the conditions of (Wendell & Hurter Jr, 1976, Corollary 2) are satisfied for problem (P2). In (Wendell & Hurter Jr, 1976, Section 2), the optimization problem

$$\inf \quad h(u) + \langle g(u), v \rangle$$
$$\text{s.t.} \quad u \in G, \ v \in H$$

is studied (Wendell & Hurter Jr, 1976, Equation 4), where it is assumed that $G \subset \mathbb{R}^m$ is an arbitrary subset and $H \subset \mathbb{R}^n$ takes the form of the standard polytope,

$$
H := \left\{ v \in \mathbb{R}^n \,\middle|\, \begin{array}{l} Av = b, \\ v_i \geq 0 \quad \forall i \in [n] \end{array} \right\}.
$$

Relating their problem to (P2), we can set $u = \text{vec}(C)$ and $v = \text{vec}(P)$, where vec denotes the transformation of a matrix into a vector. In addition, let $j : [K] \times [N] \to [KN]$ maps indices of $P$ to $\text{vec}(P)$. It follows that we can set $h(u) = 0$, $v_{j(k,n)} = p_{k,n}$, $g_{j(k,n)}(u) = w_n D(x_n, c_k)$, $G = \text{vec}(R^K)$, and $H = S_2$ with the appropriate choice of $A$ and $b$.

Besides what is stated in this proposition, (Wendell & Hurter Jr, 1976, Corollary 2) requires that $S_2$ is compact, which holds, and that $g(u)$ is continuous over $G$, which translates to $w_n D(x_n, c_k)$ being continuous with respect to $c_k \in R$. Given that $\phi$ in Definition 2.1 is convex and differentiable on int $\text{dom}(\phi)$, it holds that it is continuously differentiable on int $\text{dom}(\phi)$ (Rockafellar, 1970, Corollary 25.5.1), hence $w_n D(x_n, c_k)$ is continuous with respect to $c_k \in R$ (and more broadly, $f$ is continuous with respect to $C$ over $R^K$). This shows that (P2) satisfies the conditions of (Wendell & Hurter Jr, 1976, Corollary 2) for this proposition to be true. ∎

**Proof of Theorem 4.2.** From the assumptions that $(P^*, C_{P^*}^*)$ is a partial optimal solution and that $A(C_P^*)$ is a singleton, $(P^*, C_{P^*}^*)$ is CJ-local (Proposition D.3). Given that in addition all clusters are assumed to be non-empty, it will now be proven that $(P, C_P^*)$ is C-local.

From equation (4), $C_P^*$ is unique and continuous at $P$. Since $(P^*, C_{P^*}^*)$ is CJ-local, there exists an $\epsilon_1 > 0$ such that

$$
f(P^*, C_{P^*}^*) \leq f(P', C') \quad \forall P' \in S_2 \cap B(P^*, \epsilon_1) \text{ and } \forall C' \in R^K \cap B(C_{P^*}^*, \epsilon_1).
$$

Since $C_P^*$ is continuous at $P^*$, for any $\epsilon_1 > 0$ there exists an $\epsilon_2 > 0$ such that for all $P \in S_2 \cap B(P^*, \epsilon_2)$, it holds that $C_P^* \in B(C_{P^*}^*, \epsilon_1)$. Therefore, there exists an $\epsilon = \min(\epsilon_1, \epsilon_2) > 0$ such that

$$
F(P^*) = f(P^*, C_{P^*}^*) \leq f(P, C_P^*) = F(P) \quad \forall P \in S_2 \cap B(P^*, \epsilon),
$$

satisfying the condition of C-local optimality. ∎

**Proof of Theorem 4.3.** When $A(C_P^*)$ consists of multiple elements, there exists a point $x_g$, assigned to a cluster $a$, and a different cluster $b$ such that $D(x_g, (c_P^*)_a) = D(x_g, (c_P^*)_b)$, with $x_g \neq (c_P^*)_a$ and $x_g \neq (c_P^*)_b$, since all cluster centers are distinct by assumption. By moving $x_g$ from cluster $a$ to cluster $b$ by an amount $\alpha$ ($0 < \alpha \leq 1$), the difference in the clustering loss equals

$$
\begin{aligned}
&F(P^{\text{new}}) - F(P) \\
&= \alpha w_g (D(x_g, (c_P^*)_b) - D(x_g, (c_P^*)_a)) \\
&\quad - ((s_a(P) - \alpha w_g) D((c_{P^{\text{new}}}^*)_a, (c_P^*)_a) + (s_b(P) + \alpha w_g) D((c_{P^{\text{new}}}^*)_b, (c_P^*)_b))
\end{aligned}
$$

from Lemma 4.1. Since $x_g \neq (c_P^*)_a$, Proposition 2.2 implies that $s_a(P) - \alpha w_g > 0$. From Lemma D.2, $(c_{P^{\text{new}}}^*)_a$ and $(c_{P^{\text{new}}}^*)_b$ can be written as

$$
(c_{P^{\text{new}}}^*)_a = (c_P^*)_a - \frac{\alpha w_g (x_g - (c_P^*)_a)}{s_a(P) - \alpha w_g} \text{ and } (c_{P^{\text{new}}}^*)_b = (c_P^*)_b + \frac{\alpha w_g (x_g - (c_P^*)_b)}{s_b(P) + \alpha w_g}.
$$

Therefore, $(c_{P^{\text{new}}}^*)_a \neq (c_P^*)_a$ and $(c_{P^{\text{new}}}^*)_b \neq (c_P^*)_b$ holds, which leads to $D((c_{P^{\text{new}}}^*)_a, (c_P^*)_a) > 0$ and $D((c_{P^{\text{new}}}^*)_b, (c_P^*)_b) > 0$ from Lemma D.1. Thus,

$$
(s_a(P) - \alpha w_g) D((c_{P^{\text{new}}}^*)_a, (c_P^*)_a) + (s_b(P) + \alpha w_g) D((c_{P^{\text{new}}}^*)_b, (c_P^*)_b) > 0.
$$

From this, we have

$$
F(P^{\text{new}}) - F(P) < 0, \tag{11}
$$

which implies that $F(P)$ is not a C-local solution. Taking the contrapositive, if $F(P)$ is a C-local solution, $A(C_P^*)$ must consist of just a single element. In addition, choosing $\alpha = 1$ results in the cluster assignment $P^{\text{new}}$ being an element of $A(C_P^*) \cap T(P)$, thus, from (11), transitioning to any cluster assignment in $A(C_P^*) \cap T(P)$ guarantees a decrease in the clustering loss. ∎

**D.4. Proof of Theorem 4.4 & Theorem 4.5**

After proving the following lemma which is necessary for Theorem 4.4, we will prove Theorem 4.4 and Theorem 4.5.

**Lemma D.4.** *The LO-K-means algorithm (Algorithm 1) enters the branch in Line 9 within a finite number of iterations, and at that point, $(P, C)$ is a partial optimal solution satisfying (7) with distinct cluster centers.*

*Proof.* We first prove that the algorithm enters the branch in Line 9 in a finite number of steps.

**Case 1:** If $P$ is updated in Line 7, from Proposition B.1 with $\alpha = 1$, the clustering loss strictly decreases after the cluster centers are updated on Line 8.

When $P$ is not updated in Line 7, let the cluster assignment and the cluster centers after the $i$-th iteration ($i \geq 1$) be denoted as $P^{(i)}$ and $C^{(i)}$, respectively.

**Case 2:** If $P^{(i+1)} = P^{(i)}$, the algorithm enters the branch in Line 9, so assume that $P^{(i+1)} \neq P^{(i)}$.

**Case 3:** If $C^{(i+1)} = C^{(i)}$, then $P^{(i+2)} = P^{(i+1)}$, and the algorithm enters the branch in Line 9, given that the selection of $P$ is a deterministic function of $C$.

**Case 4:** If $P^{(i+1)} \neq P^{(i)}$ and $C^{(i+1)} \neq C^{(i)}$, given that $C^{(i+1)}$ is the unique minimizer of $f(P^{(i+1)}, \cdot)$ from Proposition 2.2, it holds that $F(P^{(i+1)}) < F(P^{(i)})$. Given that the feasible solutions in $S_1$ are finite, with a total of $K^N$ possible assignments, and that the algorithm cannot return to a previous value of $P$, due to $F$ strictly monotonically decreasing, the algorithm can only be in this state for a finite number of iterations before ultimately satisfying Cases 2 or 3 and entering the branch in Line 9. The same holds for Case 1, with the algorithm entering the branch in Line 9 after a finite number of iterations.

The algorithm enters the branch in Line 9 for Cases 2 or 3, which we now verify is with a partial optimal solution following Definition 2.9. For Case 2, given that $P^{(i+1)} = P^{(i)}$, it follows that $C^{(i+1)} = C^*_{P^{(i+1)}} = C^{(i)}$, and from Line 5 of Algorithm 1, $f(P^{(i+1)}, C^{(i)}) \leq f(P, C^{(i)}) \; \forall P \in S_2$, showing that inequality (7) holds. For Case 3, the same argument can be made given that $P^{(i+2)} = P^{(i+1)}$.

We finally prove that, when entering the branch in Line 9, the cluster centers are distinct. Assume that the branch in Line 9 is entered at iteration $i$, such that $P^{(i)} = P^{(i-1)}$ and $C^{(i)} = C^{(i-1)}$. Suppose that $C^{(i-1)}$ contains identical centers $c_a^{(i-1)}$ and $c_b^{(i-1)}$ ($a < b$). In Step 2 the cluster assignment $P^{(i)}$ for each $n$ is determined as $\min\left(\arg\min_{k' \in [K]} D(x_n, c_{k'})\right)$, resulting in cluster $b$ becoming empty. This implies that Case 1 will occur, contradicting that the algorithm entered the branch in Line 9 at iteration $i$. ∎

*Proof of Theorem 4.4.* By Lemma D.4, the branch at Line 9 is guaranteed to be entered within a finite number of iterations, and at that point, $(P, C)$ is a partial optimal solution with distinct cluster centers. Further, by Line 7 all clusters are non-empty.

For Case 1, if Function 1 updates the cluster assignment to a new value $P'$, it follows that $A(C)$ was not a singleton and from Theorem 4.3 the clustering loss strictly decreases, $F(P') < F(P)$. Since $S_1$ is finite and cluster assignments cannot return to previous values of $P$, the algorithm can only enter the branch in Line 9 and Function 1 can only improve the solution a finite number of times, hence the algorithm must converge after a finite number of iterations. At convergence, the cluster assignment is not updated in Function 1. Thus, from Theorem 4.3, $A(C)$ is unique, and by Theorem 4.2, Algorithm 1 has converged to a C-local solution. Furthermore, this also implies that the solution is CJ-local by Proposition 2.8.

For Case 2, when Function 2 updates the cluster assignment, the clustering loss strictly decreases, ensuring convergence after a finite number of iterations following the same reasoning given for Case 1. In Function 2, the clustering loss of cluster assignment $P$ is compared with all of the values of its adjacent elements. As a result, Algorithm 1 converges to a D-local solution. This solution is also C-local and CJ-local by Propositions 2.5 and 2.8. ∎

*Proof of Theorem 4.5.*

**Algorithm 1 Line 7:** For a given empty cluster $a$, finding a valid point requires searching over the $N$ points, and for each point $g$ and its cluster $b$, checking if $s_b(P) > w_g$ and for any $i \in [d]$, if $x_g[i] \neq c_b[i]$. A valid point $g$ is then transferred from its cluster $b$ to $a$, with updates $s_a(P) = s_a(P) + w_g$ and $s_b(P) = s_b(P) - w_g$.

A valid point will be found after checking at most $K$ points, resulting in a time complexity of $O(Kd)$ for each empty cluster since a point can belong to at most $K - 1$ clusters given that there exists at least 1 empty cluster. After checking $K$ points, at least two of the observed points must be assigned to the same cluster $b$. The cluster $b$ must then contain at least 2 points, hence $s_b(P) > w_g$ for all points $g$ assigned to cluster $b$. Given that all points in $X$ are unique, $x_g = c_b$ can only hold for at most one of the observed points belonging to cluster $b$, hence after checking $K$ points, at least one must be valid.

Given that there could be up to $K - 1$ empty clusters, the total time complexity is $O(K^2 d) = O(NKd) = O(NK\Gamma_\phi(d))$ given that $N > K$ by assumption.

**Recalculating $c_{k_1}$ and $c_{k_2}$ in Functions 1 and 2:** In both functions, the cluster assignments are only changed by one entry. Following Lemma D.2, the new optimal center for cluster $k \in \{k_1, k_2\}$ can be computed as $c_k^{\text{new}} = c_k + \frac{(p_{k,n}^{\text{new}} - p_{k,n})w_n(x_n - c_k)}{s_k(P) + (p_{k,n}^{\text{new}} - p_{k,n})w_n}$, i.e. $c_{k_1}^{\text{new}} = c_{k_1} - \frac{w_n(x_n - c_{k_1})}{s_{k_1}(P) - w_n}$ and $c_{k_2}^{\text{new}} = c_{k_2} + \frac{w_n(x_n - c_{k_2})}{s_{k_2}(P) + w_n}$. Given that $s_k(P)$ has been precomputed in Algorithm 1 at Line 5, the time complexity of updating $c_k$ is $O(d)$.

When studying the time complexity of Functions 1 and 2 we consider the case where for $n \in [N]$ $\min(\arg\min_{k' \in [K]} D(x_n, c_{k'}))$ and $\max(\arg\max_{k' \in [K]} D(x_n, c_{k'}))$ (assumed to also be computed in Algorithm 1 at Line 5 for Case 1) are stored in memory, or when $\arg\min_{k' \in [K]} D(x_n, c_{k'})$ for $n \in N$ needs to be recomputed.

Given that Algorithm 1 will only enter Functions 1 or 2 if $P$ has not changed values, this implies that on Line 8 the centers will not have been changed, making the use of stored values of $\arg\min_{k' \in [K]} D(x_n, c_{k'})$ in Functions 1 and 2 valid. In addition, storing these values only requires $O(N)$ of memory, which does not increase the space complexity of Algorithm 1.

**Time Complexity of Function 1:** Assuming for $n \in [N]$ the values of $\min(\arg\min_{k' \in [K]} D(x_n, c_{k'}))$ and $\max(\arg\max_{k' \in [K]} D(x_n, c_{k'}))$ have been stored in memory, at Line 3, checking if $\min(\arg\min_{k' \in [K]} D(x_n, c_{k'})) = \max(\arg\min_{k' \in [K]} D(x_n, c_{k'}))$ for $n \in [N]$ is $O(N)$. Since Line 7 can be computed in $O(d)$, the overall time complexity is $O(N + d)$. If the values of $\arg\min_{k' \in [K]} D(x_n, c_{k'})$ must be recomputed, at Line 3, for each $n \in [N]$, we must compute the function $D$ $K$ times, resulting in a total time complexity of $O(NK\Gamma_\phi(d))$.

**Time Complexity of Functions 2 and 3:** For Function 2, assuming for $n \in [N]$ the value of $\min(\arg\min_{k' \in [K]} D(x_n, c_{k'}))$ has been stored in memory, the time complexity of Line 3 equals $O(N)$. Otherwise, computing $\min(\arg\min_{k' \in [K]} D(x_n, c_{k'}))$ for all $n \in [N]$ is $O(NK\Gamma_\phi(d))$. Line 5 must be computed $O(NK)$ times. Examining (8), the time complexity of computing $\Delta_1(n, k_1, k_2)$ is $O(d + \Gamma_\phi(d)) = O(\Gamma_\phi(d))$ due to computing the new centers and $D$, hence the total time complexity of Line 5, and Function 2 in total, is $O(NK\Gamma_\phi(d))$. For Function 3, the same arguments can be applied to conclude that its time complexity is also $O(NK\Gamma_\phi(d))$.

■

# E. Experimental Details

### E.1. Details of the K-means Algorithm and Min-D-LO

In the experiments, we used the following implementation of the K-means algorithm in Algorithm 2, which is the LO-K-means algorithm excluding the new step. Its solution is guaranteed to have no empty clusters, with all cluster centers being distinct, as proven in Lemma D.4.

Function 3 gives a detailed implementation of Min-D-LO as described in Section 4.3. In particular, this function can be called instead of Function 2 in Algorithm 1, where instead of exiting after finding the first adjacent point which guarantees a clustering loss improvement, Min-D-LO finds the adjacent point which minimizes the clustering loss, while still guaranteeing convergence to a D-local solution.

---

**Algorithm 2** K-means Algorithm

---

**Input:** $X = \{x_n\}_{n \in [N]} \subset \text{int dom}(\phi) \subseteq \mathbb{R}^d$, $W = \{w_n\}_{n \in [N]} \subset \mathbb{R}_{++}$, number of clusters $K \in \mathbb{N}$

1: Sample without replacement $\{c_k\}_{k \in [K]} \subset \{x_n\}_{n \in [N]}$                                                  **(Step 1)**

2: Initialize $p_{k,n} \leftarrow 0$ for all $k \in [K]$, $n \in [N]$

3: **while** $P$ continues to change **do**

4:     $p_{k,n} \leftarrow 0$ for all $k \in [K]$, $n \in [N]$

5:     $p_{k,n} \leftarrow 1$ for all $n \in [N]$, $k = \min(\arg\min_{k' \in [K]} D(x_n, c_{k'}))$                           **(Step 2)**

6:     **if** an empty cluster $a \in [K]$ exists **then**

7:         Move a point $g \in [N]$ from a cluster $b \in [K]$ to $a$, such that $s_b(P) > w_g$ and $x_g \neq c_b$.

8:     $c_k \leftarrow \frac{\sum_{n=1}^{N} p_{k,n} w_n x_n}{\sum_{n=1}^{N} p_{k,n} w_n}$ for all $k \in [K]$                                             **(Step 3)**

**Output:** $P$: cluster assignment, $C$: cluster centers

---

**Function 3** Variant of Function 2 Minimizing $\Delta_1$

---

1: **function** MIN-D-LO$(X, W, P, C, D)$

2:     $\Delta_{\min} = 0$

3:     $(n^m, k_1^m, k_2^m) = (0, 0, 0)$

4:     **for** $n = 1, 2, \ldots, N$ **do**

5:         $k_1 \leftarrow \min(\arg\min_{k' \in [K]} D(x_n, c_{k'}))$

6:         **for** $k_2 = 1, 2, \ldots, k_1 - 1, k_1 + 1, \ldots, K$ **do**

7:             **if** $\Delta_1(n, k_1, k_2) < \Delta_{\min}$ **then**

8:                 $\Delta_{\min} = \Delta_1(n, k_1, k_2)$

9:                 $(n^m, k_1^m, k_2^m) = (n, k_1, k_2)$

10:    **if** $\Delta_{\min} < 0$ **then**

11:       $p_{k_2^m, n^m} \leftarrow 1$

12:       Recalculate $c_{k_2^m}$

13:       **if** $s_{k_1^m}(P) = w_{n^m}$ **then**

14:          $p_{k_1^m, n^m} \leftarrow 0$

15:       **else**

16:          $p_{k_1^m, n^m} \leftarrow 0$

17:          Recalculate $c_{k_1^m}$

---

### E.2. Details of the Real-World Datasets

We conduct our experiments on the following five datasets.

- **Iris (Fisher, 1936)**: This dataset consists of 150 instances and 4 features, where each instance represents a plant.

- **Wine Quality (Cortez et al., 2009)**: A dataset with 6,497 instances and 11 features, where each instance corresponds to a wine sample with quality ratings.

- **Yeast (Nakai & Kanehisa, 1991; 1992)**: This dataset contains 1,484 instances and 8 features, where each instance represents a protein sample with attributes related to its cellular localization.

- **Predict Students' Dropout and Academic Success (Martins et al., 2021)**: This dataset consists of 4,424 instances and 36 features related to students' academic performance and dropout risk.

- **News20 (scikit-learn developers, 2017)**: This dataset contains 11,314 instances and 131,017 features, representing word frequencies in news articles. Since both the number of instances and features are large, experiments were conducted in the following two cases.

  1. Using the first 2,000 instances, focusing on the 1,089 features with word frequencies between 2% and 80%.
  2. Using the first 200 instances and 131,017 features.

# F. Additional Experiments

This section presents experimental results not covered in Section 5.

### F.1. Synthetic Datasets

Recall that synthetic datasets were generated by uniformly sampling $N$ data points from the space $[1, 10]^d$, restricted to integer values. If the same point is selected multiple times, the number of times it is sampled is assigned as its weight.

Figures 4 and 5 present the percentage increase in the number of iterations and the number of times the new step (Lines 9–11) was invoked, respectively, when there is an improvement in the clustering loss using C-LO instead of K-means. As seen from Figure 5, the new step (Lines 9–11) only needs to be called once in most cases.

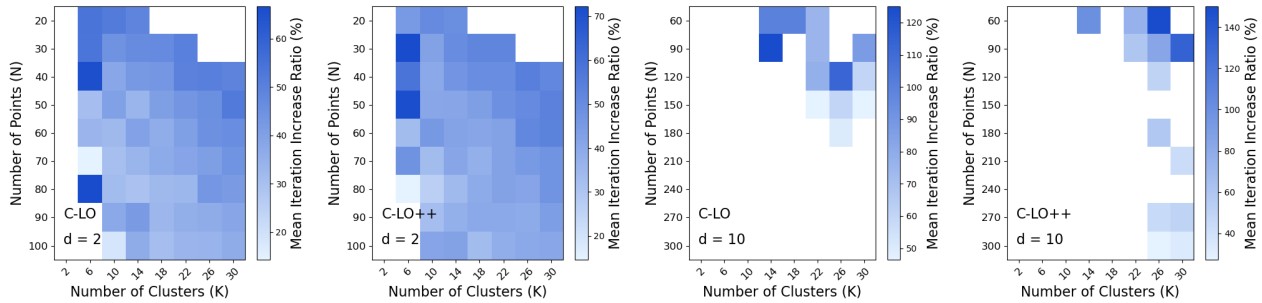

*Figure 4.* The average rate of increase in the number of iterations when C-LO improves over K-means across two different initialization methods and dimensions, with $D$ equal to the squared Euclidean distance. Each $N, K$ cell represents the results from 1,000 runs of both algorithms. The ratio is given by $(I_{\text{LO}} - I)/I$, where $I_{\text{LO}}$ is the number of iterations using C-LO and $I$ is the number of iterations using K-means. Darker colors indicate a higher percentage increase in iterations.

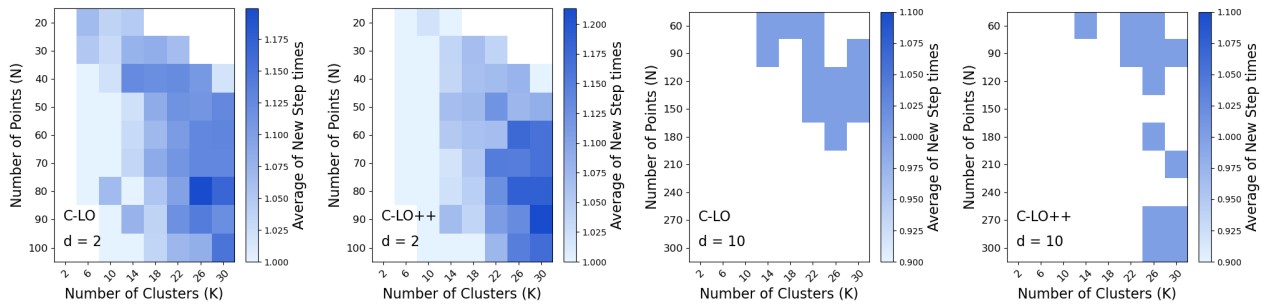

*Figure 5.* The average number of times the new step was invoked when C-LO improved over K-means across two different initialization methods and dimensions, with $D$ equal to the squared Euclidean distance. Each $N, K$ cell represents the results from 1,000 runs of all algorithms. Darker colors indicate a higher frequency of new step invocations.

Figures 6–9 show the experimental results for D-LO. These results indicate that D-LO improves the clustering loss in many cases, regardless of the dimensionality. When the number of clusters $K$ is greater than 5, the clustering loss decreases in most cases. The percentage decrease in the clustering loss and the increase in iterations tend to be higher when the number of clusters $K$ is large relative to the number of data points $N$ (Figures 7 and 8). Additionally, the number of times the new step is invoked increases when both $N$ and $K$ are large (Figure 9).

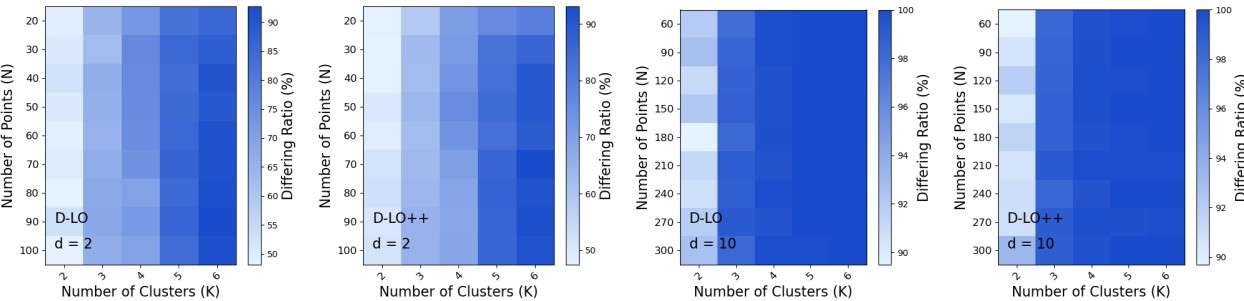

*Figure 6.* The proportion of cases where the clustering loss is improved over K-means by using D-LO across two different initialization methods and dimensions, with $D$ equal to the squared Euclidean distance. Each $N, K$ cell represents the results from 1,000 runs of both algorithms. Darker colors indicate a higher frequency of clustering loss improvement.

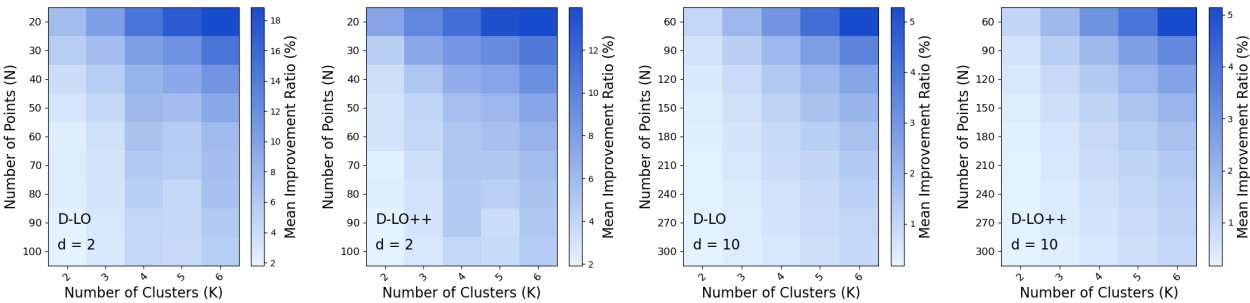

*Figure 7.* The average improvement rate of the clustering loss when D-LO improves over K-means across two different initialization methods and dimensions, with $D$ equal to the squared Euclidean distance. Each $N, K$ cell represents the results from 1,000 runs of both algorithms. The ratio is given by $(F(P) - F(P_{\text{LO}}))/F(P)$, where $P_{\text{LO}}$ is the output of D-LO and $P$ is the output of K-means. Darker colors indicate a higher percentage of clustering loss improvement.

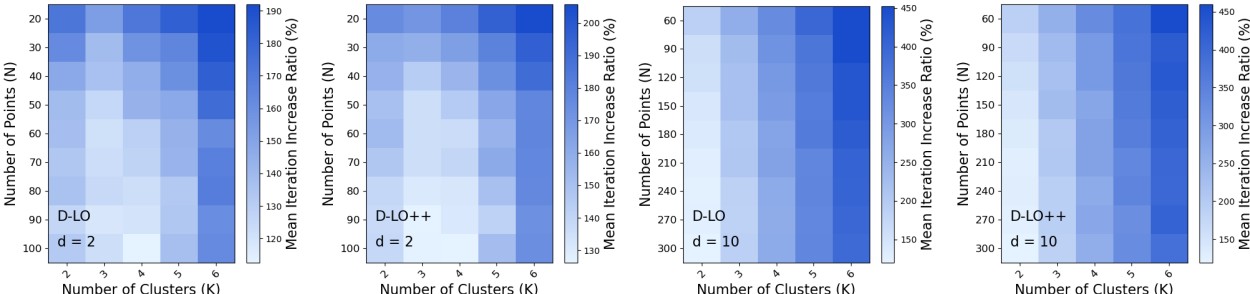

*Figure 8.* The average rate of increase in the number of iterations when D-LO improves over K-means across two different initialization methods and dimensions, with $D$ equal to the squared Euclidean distance. Each $N, K$ cell represents the results from 1,000 runs of both algorithms. The ratio is given by $(I_{\text{LO}} - I)/I$, where $I_{\text{LO}}$ is the number of iterations using D-LO and $I$ is the number of iterations using K-means. Darker colors indicate a higher percentage increase in iterations.

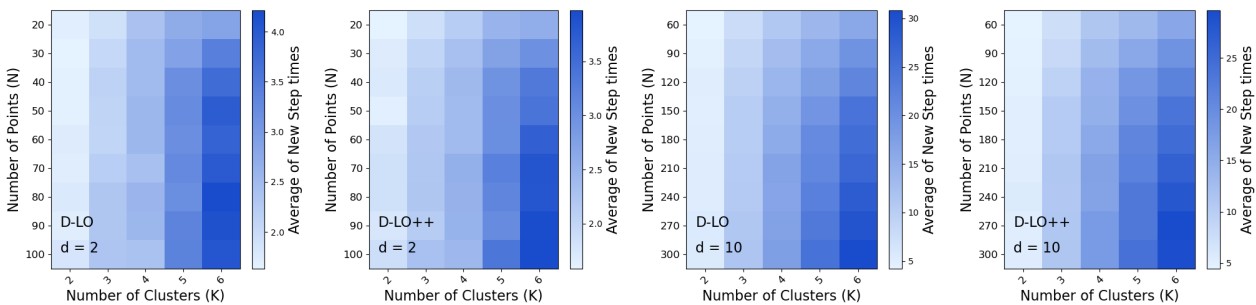

*Figure 9.* The average number of times the new step was invoked when D-LO improves over K-means across two different initialization methods and dimensions, with $D$ equal to the squared Euclidean distance. Each $N, K$ cell represents the results from 1,000 runs of all algorithms. Darker colors indicate a higher frequency of new step invocations.

### F.2. Real-World Datasets

In this subsection, we present the results for real-world datasets that were not covered in Section 5.3.

As shown in the following tables, the clustering loss of C-LO is identical to that of K-means in all cases. These findings suggest that although the K-means algorithm was not explicitly designed to guarantee this property, it still converges to a C-local solution in most real-world datasets. However, by comparing the computation time of K-means with C-LO, it is observed that the time to verify that the K-means algorithm has converged to a C-local solution using C-LO is negligible.

Across all datasets, D-LO and Min-D-LO consistently outperform K-means in terms of both the mean and the minimum of the clustering loss, regardless of the initialization method. Notably, stronger improvements were observed in high-dimensional datasets such as in Tables 10 and 11.

*Table 2.* Iris dataset ($N = 150, d = 4$): Mean, variance, and minimum of the clustering loss, along with the average computation time and average number of iterations over 20 runs for each initialization method and number of clusters, for K-means, C-LO, D-LO, and Min-D-LO with $D$ chosen as the squared Euclidean distance.

| | Initialization | Random | | | | K-means++ | | | |
|---|---|---|---|---|---|---|---|---|---|
| $K$ | Algorithm | Mean $\pm$ Variance | Minimum | Time(s) | Num Iter | Mean $\pm$ Variance | Minimum | Time(s) | Num Iter |
| 5 | K-means | $57.54 \pm 9.78$ | **46.54** | $< 0.001$ | 9 | $50.58 \pm 4.74$ | **46.54** | $< 0.001$ | 8 |
| | C-LO | $57.54 \pm 9.78$ | **46.54** | $< 0.001$ | 9 | $50.58 \pm 4.74$ | **46.54** | $< 0.001$ | 8 |
| | D-LO | **$57.32 \pm 9.83$** | **46.54** | $< 0.001$ | 13 | **$50.30 \pm 4.70$** | **46.54** | $< 0.001$ | 13 |
| | Min-D-LO | **$57.32 \pm 9.83$** | **46.54** | $< 0.001$ | 13 | **$50.30 \pm 4.70$** | **46.54** | $< 0.001$ | 13 |
| 10 | K-means | $31.55 \pm 5.42$ | 26.78 | $< 0.001$ | 8 | $29.57 \pm 2.97$ | 26.01 | $< 0.001$ | 7 |
| | C-LO | $31.55 \pm 5.42$ | 26.78 | $< 0.001$ | 8 | $29.57 \pm 2.97$ | 26.01 | $< 0.001$ | 7 |
| | D-LO | **$30.53 \pm 5.00$** | **26.18** | $< 0.001$ | 20 | **$28.92 \pm 3.00$** | **25.94** | $< 0.001$ | 17 |
| | Min-D-LO | $30.55 \pm 4.99$ | **26.18** | $< 0.001$ | 19 | $28.93 \pm 3.00$ | **25.94** | $< 0.001$ | 17 |
| 25 | K-means | $15.98 \pm 1.64$ | 13.67 | $< 0.001$ | 7 | $13.73 \pm 0.68$ | 12.70 | $< 0.001$ | 6 |
| | C-LO | $15.98 \pm 1.64$ | 13.67 | $< 0.001$ | 7 | $13.73 \pm 0.68$ | 12.70 | $< 0.001$ | 6 |
| | D-LO | $14.59 \pm 1.68$ | **11.89** | $< 0.001$ | 36 | **$12.58 \pm 0.45$** | **11.83** | $< 0.001$ | 31 |
| | Min-D-LO | **$14.49 \pm 1.76$** | 12.02 | $< 0.001$ | 30 | $12.61 \pm 0.43$ | 12.07 | $< 0.001$ | 27 |
| 50 | K-means | $8.65 \pm 1.27$ | 7.05 | $< 0.001$ | 5 | $6.40 \pm 0.34$ | 5.52 | $< 0.001$ | 5 |
| | C-LO | $8.65 \pm 1.27$ | 7.05 | $< 0.001$ | 5 | $6.40 \pm 0.34$ | 5.52 | $< 0.001$ | 5 |
| | D-LO | $7.13 \pm 1.50$ | 5.66 | 0.002 | 48 | **$5.36 \pm 0.24$** | **5.04** | 0.002 | 37 |
| | Min-D-LO | **$7.09 \pm 1.35$** | **5.62** | 0.002 | 39 | $5.40 \pm 0.23$ | **5.04** | 0.002 | 30 |

*Table 3.* Wine Quality dataset ($N = 6{,}497, d = 11$): Mean, variance, and minimum of the clustering loss, along with the average computation time and average number of iterations over 20 runs for each initialization method and number of clusters, for K-means, C-LO, D-LO, and Min-D-LO with $D$ chosen as the squared Euclidean distance.

| | Initialization | Random | | | | K-means++ | | | |
|---|---|---|---|---|---|---|---|---|---|
| $K$ | Algorithm | Mean $\pm$ Variance | Minimum | Time(s) | Num Iter | Mean $\pm$ Variance | Minimum | Time(s) | Num Iter |
| 5 | K-means | $2{,}393{,}869 \pm 67$ | $2{,}393{,}751$ | 0.005 | 40 | $2{,}499{,}611 \pm 211{,}445$ | $2{,}393{,}754$ | 0.005 | 40 |
| | C-LO | $2{,}393{,}869 \pm 67$ | $2{,}393{,}751$ | 0.006 | 40 | $2{,}499{,}611 \pm 211{,}445$ | $2{,}393{,}754$ | 0.005 | 40 |
| | D-LO | $\mathbf{2{,}393{,}759} \pm 12$ | $\mathbf{2{,}393{,}742}$ | 0.006 | 49 | $\mathbf{2{,}393{,}753} \pm 11$ | $\mathbf{2{,}393{,}742}$ | 0.008 | 63 |
| | Min-D-LO | $\mathbf{2{,}393{,}759} \pm 12$ | $\mathbf{2{,}393{,}742}$ | 0.006 | 49 | $\mathbf{2{,}393{,}753} \pm 11$ | $2{,}393{,}746$ | 0.008 | 62 |
| 10 | K-means | $1{,}378{,}087 \pm 6{,}569$ | $\mathbf{1{,}367{,}222}$ | 0.01 | 51 | $1{,}381{,}737 \pm 9{,}516$ | $1{,}365{,}020$ | 0.01 | 52 |
| | C-LO | $1{,}378{,}087 \pm 6{,}569$ | $\mathbf{1{,}367{,}222}$ | 0.01 | 51 | $1{,}381{,}737 \pm 9{,}516$ | $1{,}365{,}020$ | 0.01 | 52 |
| | D-LO | $\mathbf{1{,}377{,}711} \pm 6{,}571$ | $\mathbf{1{,}367{,}222}$ | 0.02 | 74 | $\mathbf{1{,}380{,}941} \pm 9{,}674$ | $\mathbf{1{,}364{,}944}$ | 0.02 | 87 |
| | Min-D-LO | $\mathbf{1{,}377{,}711} \pm 6{,}571$ | $\mathbf{1{,}367{,}222}$ | 0.02 | 74 | $\mathbf{1{,}380{,}941} \pm 9{,}674$ | $\mathbf{1{,}364{,}944}$ | 0.02 | 84 |
| 25 | K-means | $681{,}397 \pm 24{,}123$ | $659{,}902$ | 0.04 | 86 | $654{,}219 \pm 15{,}173$ | $631{,}244$ | 0.03 | 52 |
| | C-LO | $681{,}397 \pm 24{,}123$ | $659{,}902$ | 0.04 | 86 | $654{,}219 \pm 15{,}173$ | $631{,}244$ | 0.02 | 52 |
| | D-LO | $\mathbf{665{,}882} \pm 7{,}059$ | $\mathbf{654{,}126}$ | 0.12 | 235 | $649{,}046 \pm 14{,}736$ | $\mathbf{630{,}546}$ | 0.06 | 121 |
| | Min-D-LO | $666{,}641 \pm 7{,}082$ | $\mathbf{654{,}126}$ | 0.10 | 202 | $\mathbf{649{,}001} \pm 14{,}774$ | $631{,}157$ | 0.05 | 107 |
| 50 | K-means | $434{,}056 \pm 14{,}094$ | $402{,}580$ | 0.11 | 63 | $376{,}544 \pm 8{,}469$ | $367{,}108$ | 0.05 | 45 |
| | C-LO | $434{,}056 \pm 14{,}094$ | $402{,}580$ | 0.08 | 63 | $376{,}544 \pm 8{,}469$ | $367{,}108$ | 0.06 | 45 |
| | D-LO | $\mathbf{424{,}122} \pm 20{,}475$ | $\mathbf{381{,}856}$ | 0.28 | 259 | $\mathbf{372{,}716} \pm 5{,}701$ | $365{,}447$ | 0.21 | 188 |
| | Min-D-LO | $424{,}435 \pm 20{,}610$ | $382{,}197$ | 0.29 | 229 | $373{,}075 \pm 5{,}877$ | $\mathbf{364{,}596}$ | 0.18 | 164 |

*Table 4.* Yeast dataset ($N = 1{,}484, d = 8$): Mean, variance, and minimum of the clustering loss, along with the average computation time and average number of iterations over 20 runs for each initialization method and number of clusters, for K-means, C-LO, D-LO, and Min-D-LO with $D$ chosen as the squared Euclidean distance.

| | Initialization | Random | | | | K-means++ | | | |
|---|---|---|---|---|---|---|---|---|---|
| $K$ | Algorithm | Mean $\pm$ Variance | Minimum | Time(s) | Num Iter | Mean $\pm$ Variance | Minimum | Time(s) | Num Iter |
| 5 | K-means | $64.1811 \pm 0.5625$ | $\mathbf{63.3292}$ | $< 0.001$ | 27 | $64.7247 \pm 1.8557$ | $\mathbf{63.3292}$ | $< 0.001$ | 19 |
| | C-LO | $64.1811 \pm 0.5625$ | $\mathbf{63.3292}$ | $< 0.001$ | 27 | $64.7247 \pm 1.8557$ | $\mathbf{63.3292}$ | $< 0.001$ | 19 |
| | D-LO | $64.0656 \pm 0.2458$ | $\mathbf{63.3292}$ | $< 0.001$ | 31 | $64.7215 \pm 1.8564$ | $\mathbf{63.3292}$ | $< 0.001$ | 21 |
| | Min-D-LO | $\mathbf{64.0655} \pm 0.2457$ | $\mathbf{63.3292}$ | $< 0.001$ | 31 | $\mathbf{64.7214} \pm 1.8564$ | $\mathbf{63.3292}$ | $< 0.001$ | 21 |
| 10 | K-means | $49.7885 \pm 2.9719$ | $45.9045$ | 0.001 | 28 | $47.4975 \pm 2.7205$ | $45.3131$ | 0.001 | 28 |
| | C-LO | $49.7885 \pm 2.9719$ | $45.9045$ | 0.001 | 28 | $47.4975 \pm 2.7205$ | $45.3131$ | 0.001 | 28 |
| | D-LO | $\mathbf{48.9709} \pm 3.0474$ | $\mathbf{45.7640}$ | 0.003 | 68 | $\mathbf{47.4400} \pm 2.6732$ | $\mathbf{45.3040}$ | 0.003 | 56 |
| | Min-D-LO | $48.9734 \pm 3.0456$ | $\mathbf{45.7640}$ | 0.003 | 61 | $47.4409 \pm 2.6750$ | $\mathbf{45.3040}$ | 0.003 | 52 |
| 25 | K-means | $31.9759 \pm 1.0614$ | $30.2560$ | 0.003 | 31 | $31.4602 \pm 0.9611$ | $30.3055$ | 0.003 | 29 |
| | C-LO | $31.9759 \pm 1.0614$ | $30.2560$ | 0.003 | 31 | $31.4602 \pm 0.9611$ | $30.3055$ | 0.003 | 9 |
| | D-LO | $31.3294 \pm 1.0120$ | $30.1804$ | 0.02 | 164 | $\mathbf{31.1418} \pm 0.9025$ | $\mathbf{30.0612}$ | 0.02 | 157 |
| | Min-D-LO | $\mathbf{31.3285} \pm 1.0552$ | $\mathbf{30.1802}$ | 0.02 | 131 | $31.1563 \pm 0.8695$ | $30.1331$ | 0.02 | 126 |
| 50 | K-means | $23.4076 \pm 0.2833$ | $22.7046$ | 0.005 | 26 | $22.9436 \pm 0.2097$ | $22.5824$ | 0.005 | 24 |
| | C-LO | $23.4076 \pm 0.2833$ | $22.7046$ | 0.006 | 26 | $22.9436 \pm 0.2097$ | $22.5824$ | 0.006 | 24 |
| | D-LO | $22.8217 \pm 0.2970$ | $22.2496$ | 0.07 | 286 | $\mathbf{22.4312} \pm 0.1982$ | $\mathbf{22.0767}$ | 0.06 | 257 |
| | Min-D-LO | $\mathbf{22.7697} \pm 0.2797$ | $\mathbf{22.2465}$ | 0.06 | 221 | $22.4631 \pm 0.1899$ | $22.1379$ | 0.05 | 182 |

*Table 5.* Predict Students' Dropout and Academic Success dataset ($N = 4{,}424, d = 36$): Mean, variance, and minimum of the clustering loss, along with the average computation time and average number of iterations over 20 runs for each initialization method and number of clusters, for K-means, C-LO, D-LO, and Min-D-LO with $D$ chosen as the squared Euclidean distance.

| | Initialization | Random | | | | K-means++ | | | |
|---|---|---|---|---|---|---|---|---|---|
| $K$ | Algorithm | Mean $\pm$ Variance | Minimum | Time(s) | Num Iter | Mean $\pm$ Variance | Minimum | Time(s) | Num Iter |
| 5 | K-means | $153{,}167{,}019 \pm 110{,}756{,}304$ | **38,258,341** | 0.002 | 7 | $\mathbf{44{,}852{,}897} \pm 13{,}571{,}603$ | **38,258,341** | 0.001 | 4 |
| | C-LO | $153{,}167{,}019 \pm 110{,}756{,}304$ | **38,258,341** | 0.002 | 7 | $\mathbf{44{,}852{,}897} \pm 13{,}571{,}603$ | **38,258,341** | 0.001 | 4 |
| | D-LO | $\mathbf{153{,}166{,}788} \pm 110{,}756{,}446$ | **38,258,341** | 0.003 | 8 | $\mathbf{44{,}852{,}897} \pm 13{,}571{,}603$ | **38,258,341** | 0.001 | 4 |
| | Min-D-LO | $\mathbf{153{,}166{,}788} \pm 110{,}756{,}446$ | **38,258,341** | 0.003 | 8 | $\mathbf{44{,}852{,}897} \pm 13{,}571{,}603$ | **38,258,341** | 0.001 | 4 |
| 10 | K-means | $18{,}215{,}696 \pm 3{,}940{,}334$ | **13,417,982** | 0.008 | 13 | $14{,}799{,}899 \pm 1{,}951{,}762$ | **11,998,592** | 0.005 | 9 |
| | C-LO | $18{,}215{,}696 \pm 3{,}940{,}334$ | **13,417,982** | 0.008 | 13 | $14{,}799{,}899 \pm 1{,}951{,}762$ | **11,998,592** | 0.005 | 9 |
| | D-LO | $\mathbf{18{,}149{,}464} \pm 3{,}957{,}643$ | **13,417,982** | 0.01 | 16 | $14{,}770{,}849 \pm 1{,}956{,}191$ | **11,998,592** | 0.007 | 12 |
| | Min-D-LO | $\mathbf{18{,}149{,}464} \pm 3{,}957{,}643$ | **13,417,982** | 0.01 | 16 | $\mathbf{14{,}770{,}794} \pm 1{,}956{,}074$ | **11,998,592** | 0.007 | 12 |
| 25 | K-means | $7{,}892{,}024 \pm 1{,}036{,}157$ | 7,037,040 | 0.03 | 21 | $7{,}129{,}832 \pm 331{,}980$ | 6,652,215 | 0.03 | 20 |
| | C-LO | $7{,}892{,}024 \pm 1{,}036{,}157$ | 7,037,040 | 0.03 | 21 | $7{,}129{,}832 \pm 331{,}980$ | 6,652,215 | 0.03 | 20 |
| | D-LO | $\mathbf{7{,}852{,}829} \pm 1{,}032{,}327$ | **7,036,545** | 0.08 | 57 | $\mathbf{7{,}054{,}448} \pm 272{,}093$ | **6,648,003** | 0.06 | 40 |
| | Min-D-LO | $7{,}854{,}002 \pm 1{,}031{,}819$ | 7,036,990 | 0.08 | 55 | $7{,}055{,}679 \pm 274{,}184$ | **6,648,003** | 0.06 | 38 |
| 50 | K-means | $5{,}476{,}986 \pm 298{,}309$ | 5,053,050 | 0.07 | 26 | $5{,}020{,}874 \pm 148{,}015$ | 4,796,299 | 0.07 | 25 |
| | C-LO | $5{,}476{,}986 \pm 298{,}309$ | 5,053,050 | 0.07 | 26 | $5{,}020{,}874 \pm 148{,}015$ | 4,796,299 | 0.07 | 25 |
| | D-LO | $\mathbf{5{,}418{,}530} \pm 299{,}341$ | 5,045,559 | 0.32 | 113 | $\mathbf{4{,}982{,}183} \pm 132{,}509$ | **4,787,903** | 0.28 | 99 |
| | Min-D-LO | $5{,}419{,}034 \pm 294{,}567$ | **4,957,810** | 0.31 | 106 | $4{,}983{,}925 \pm 135{,}801$ | 4,788,567 | 0.25 | 86 |

## F.3. Experiments with Other Dissimilarity Measures (KL Divergence & Itakura-Saito Divergence)

We also conducted experiments using dissimilarity measures $D$ other than the squared Euclidean distance.

Tables 6–9 present the experimental results for the Iris and Yeast datasets. In these experiments, KL divergence was used for the results in Tables 6 and 7, while Itakura-Saito divergence was used for those in Tables 8 and 9.

The KL divergence has $\text{dom}(\phi) = \mathbb{R}^d_+$, while the Itakura-Saito divergence has $\text{dom}(\phi) = \mathbb{R}^d_{++}$. Therefore, we preprocessed the datasets accordingly, noting that neither the Iris nor Yeast dataset contains negative values. When using these divergences, dimensions that do not belong to $\text{int dom}(\phi) = \mathbb{R}^d_{++}$ were excluded from consideration. As a result, the dimensionality of the Yeast dataset was reduced from 8 to 4.

As shown in Tables 6–9, D-LO and Min-D-LO consistently reduced the clustering loss regardless of the dissimilarity measure used.

*Table 6.* Iris dataset ($N = 150, d = 4$): Mean, variance, and minimum of the clustering loss, along with the average computation time and average number of iterations over 20 runs for each number of clusters, for K-means, C-LO, D-LO, and Min-D-LO with $D$ chosen as the KL divergence.

| $K$ | Algorithm | Mean $\pm$ Variance | Minimum | Time(s) | Num Iter |
|---|---|---|---|---|---|
| 5 | K-means | $10.3353 \pm 4.3956$ | **7.3918** | $< 0.001$ | 7 |
| | C-LO | $10.3353 \pm 4.3956$ | **7.3918** | $< 0.001$ | 7 |
| | D-LO | $\mathbf{10.2722} \pm 4.4126$ | **7.3918** | 0.002 | 13 |
| | Min-D-LO | $\mathbf{10.2722} \pm 4.4126$ | **7.3918** | 0.002 | 13 |
| 10 | K-means | $5.0824 \pm 0.4933$ | 4.5094 | 0.002 | 9 |
| | C-LO | $5.0824 \pm 0.4933$ | 4.5094 | 0.002 | 9 |
| | D-LO | $\mathbf{4.9544} \pm 0.5108$ | 4.3596 | 0.006 | 22 |
| | Min-D-LO | $4.9588 \pm 0.5056$ | **4.3561** | 0.007 | 20 |
| 25 | K-means | $2.6667 \pm 0.2369$ | 2.3077 | 0.003 | 7 |
| | C-LO | $2.6667 \pm 0.2369$ | 2.3077 | 0.003 | 7 |
| | D-LO | $\mathbf{2.3793} \pm 0.1636$ | 2.1739 | 0.03 | 44 |
| | Min-D-LO | $2.3917 \pm 0.1726$ | **2.1579** | 0.04 | 36 |
| 50 | K-means | $1.4186 \pm 0.1209$ | 1.2542 | 0.005 | 5 |
| | C-LO | $1.4186 \pm 0.1209$ | 1.2542 | 0.005 | 5 |
| | D-LO | $\mathbf{1.1139} \pm 0.1230$ | **0.9836** | 0.09 | 63 |
| | Min-D-LO | $1.1260 \pm 0.1274$ | 1.0009 | 0.12 | 51 |

*Table 7.* Yeast dataset ($N = 1,484, d = 4$): Mean, variance, and minimum of the clustering loss, along with the average computation time and average number of iterations over 20 runs for each number of clusters, for K-means, C-LO, D-LO, and Min-D-LO with $D$ chosen as the KL divergence.

| $K$ | Algorithm | Mean $\pm$ Variance | Minimum | Time(s) | Num Iter |
|---|---|---|---|---|---|
| 5 | K-means | $24.7930 \pm 0.1004$ | 24.7061 | 0.02 | 27 |
| | C-LO | $24.7930 \pm 0.1004$ | 24.7061 | 0.02 | 27 |
| | D-LO | $\mathbf{24.7698} \pm 0.0698$ | 24.7061 | 0.05 | 46 |
| | Min-D-LO | $24.7704 \pm 0.0695$ | **24.7058** | 0.05 | 43 |
| 10 | K-means | $17.1498 \pm 0.2249$ | 16.3044 | 0.05 | 33 |
| | C-LO | $17.1498 \pm 0.2249$ | 16.3044 | 0.05 | 33 |
| | D-LO | $\mathbf{17.1263} \pm 0.2197$ | **16.2683** | 0.11 | 57 |
| | Min-D-LO | $17.1273 \pm 0.2207$ | **16.2683** | 0.12 | 54 |
| 25 | K-means | $9.6594 \pm 0.6781$ | 9.0472 | 0.12 | 31 |
| | C-LO | $9.6594 \pm 0.6781$ | 9.0472 | 0.12 | 31 |
| | D-LO | $\mathbf{9.2017} \pm 0.3940$ | 8.9245 | 0.61 | 121 |
| | Min-D-LO | $9.3429 \pm 0.5815$ | **8.9205** | 0.65 | 99 |
| 50 | K-means | $5.8949 \pm 0.2300$ | 5.5487 | 0.24 | 32 |
| | C-LO | $5.8949 \pm 0.2300$ | 5.5487 | 0.24 | 32 |
| | D-LO | $\mathbf{5.6315} \pm 0.2318$ | **5.2839** | 2.34 | 213 |
| | Min-D-LO | $5.6817 \pm 0.2305$ | 5.3299 | 2.45 | 163 |

*Table 8.* Iris dataset ($N = 150, d = 4$): Mean, variance, and minimum of the clustering loss, along with the average computation time and average number of iterations over 20 runs for each number of clusters, for K-means, C-LO, D-LO, and Min-D-LO with $D$ chosen as the Itakura-Saito divergence.

| $K$ | Algorithm | Mean $\pm$ Variance | Minimum | Time(s) | Num Iter |
|---|---|---|---|---|---|
| 5 | K-means | $5.5113 \pm 1.3082$ | **3.4763** | $< 0.001$ | 7 |
| | C-LO | $5.5113 \pm 1.3082$ | **3.4763** | $< 0.001$ | 7 |
| | D-LO | $\mathbf{5.4170} \pm 1.3316$ | **3.4763** | 0.001 | 10 |
| | Min-D-LO | $\mathbf{5.4170} \pm 1.3317$ | **3.4763** | 0.001 | 10 |
| 10 | K-means | $2.2542 \pm 0.5641$ | **1.7175** | 0.001 | 9 |
| | C-LO | $2.2542 \pm 0.5641$ | **1.7175** | 0.001 | 9 |
| | D-LO | $\mathbf{2.2213} \pm 0.5613$ | **1.7175** | 0.004 | 15 |
| | Min-D-LO | $2.2222 \pm 0.5623$ | **1.7175** | 0.004 | 15 |
| 25 | K-means | $1.0272 \pm 0.1306$ | 0.7975 | 0.003 | 8 |
| | C-LO | $1.0272 \pm 0.1306$ | 0.7975 | 0.003 | 8 |
| | D-LO | $0.9031 \pm 0.0637$ | **0.7828** | 0.03 | 37 |
| | Min-D-LO | $\mathbf{0.8997} \pm 0.0686$ | **0.7828** | 0.03 | 31 |
| 50 | K-means | $0.5015 \pm 0.0544$ | 0.3934 | 0.005 | 6 |
| | C-LO | $0.5015 \pm 0.0544$ | 0.3934 | 0.005 | 6 |
| | D-LO | $0.4067 \pm 0.0352$ | **0.3341** | 0.06 | 51 |
| | Min-D-LO | $\mathbf{0.4063} \pm 0.0344$ | 0.3356 | 0.08 | 45 |

*Table 9.* Yeast dataset ($N = 1,484, d = 4$): Mean, variance, and minimum of the clustering loss, along with the average computation time and average number of iterations over 20 runs for each number of clusters, for K-means, C-LO, D-LO, and Min-D-LO with $D$ chosen as the Itakura-Saito divergence.

| $K$ | Algorithm | Mean $\pm$ Variance | Minimum | Time(s) | Num Iter |
|---|---|---|---|---|---|
| 5 | K-means | $52.6593 \pm 0.3565$ | $52.2224$ | 0.02 | 28 |
| | C-LO | $52.6593 \pm 0.3565$ | $52.2224$ | 0.02 | 28 |
| | D-LO | $\mathbf{52.5749} \pm 0.3242$ | $\mathbf{52.2220}$ | 0.04 | 43 |
| | Min-D-LO | $\mathbf{52.5749} \pm 0.3242$ | $\mathbf{52.2220}$ | 0.04 | 42 |
| 10 | K-means | $35.5623 \pm 0.5344$ | $34.6253$ | 0.04 | 31 |
| | C-LO | $35.5623 \pm 0.5344$ | $34.6253$ | 0.04 | 31 |
| | D-LO | $\mathbf{35.4464} \pm 0.4127$ | $34.6202$ | 0.10 | 65 |
| | Min-D-LO | $35.4539 \pm 0.4179$ | $\mathbf{34.6196}$ | 0.11 | 59 |
| 25 | K-means | $20.2199 \pm 1.0484$ | $19.1600$ | 0.09 | 30 |
| | C-LO | $20.2199 \pm 1.0484$ | $19.1600$ | 0.09 | 30 |
| | D-LO | $19.6271 \pm 0.9731$ | $18.7461$ | 0.48 | 111 |
| | Min-D-LO | $\mathbf{19.5247} \pm 0.8298$ | $\mathbf{18.7344}$ | 0.57 | 106 |
| 50 | K-means | $12.4824 \pm 0.4306$ | $11.6719$ | 0.18 | 29 |
| | C-LO | $12.4824 \pm 0.4306$ | $11.6719$ | 0.18 | 29 |
| | D-LO | $\mathbf{12.0247} \pm 0.3544$ | $\mathbf{11.2726}$ | 1.58 | 175 |
| | Min-D-LO | $12.0712 \pm 0.4092$ | $11.3139$ | 1.74 | 145 |

## F.4. Comparison with the D-local Algorithm Proposed by Peng & Xia (2005)

To compare the computation time of D-LO and Min-D-LO with the D-local algorithm proposed in (Peng & Xia, 2005, Section 3.2.1), we evaluated their algorithm, hereafter referred to as D-LO-P&X, on the most computationally challenging datasets, news20 1 & 2.

As shown in Tables 10 and 11, we observe that D-LO, Min-D-LO, and D-LO-P&X consistently achieved similar levels of clustering loss, given that they all converge to D-local solutions, while D-LO-P&X is significantly slower than all other methods (in red). This is because D-LO-P&X is not based on the K-means algorithm, but instead on directly examining whether moving to an adjacent extreme point decreases the clustering loss.

*Table 10.* News20 dataset 1 ($N = 2{,}000, d = 1{,}089$): Mean, variance, and minimum of the clustering loss, along with the average computation time and average number of iterations over 20 runs for each initialization method and number of clusters, for K-means, C-LO, D-LO, Min-D-LO, and D-LO-P&X with $D$ chosen as the squared Euclidean distance.

| | Initialization | Random | | | | K-means++ | | | |
|---|---|---|---|---|---|---|---|---|---|
| $K$ | Algorithm | Mean ± Variance | Minimum | Time(s) | Num Iter | Mean ± Variance | Minimum | Time(s) | Num Iter |
| | K-means | $984{,}655 \pm 55{,}781$ | 864,073 | 0.31 | 31 | $870{,}899 \pm 80{,}013$ | 808,450 | 0.20 | 18 |
| | C-LO | $984{,}655 \pm 55{,}781$ | 864,073 | 0.33 | 31 | $870{,}899 \pm 80{,}013$ | 808,450 | 0.18 | 18 |
| 5 | D-LO | $808{,}391 \pm 0$ | 808,391 | 1.94 | 168 | $\mathbf{806{,}236 \pm 3{,}292}$ | **801,207** | 0.82 | 74 |
| | Min-D-LO | $\mathbf{808{,}032 \pm 1{,}566}$ | **801,207** | 0.97 | 81 | $806{,}595 \pm 3{,}111$ | **801,207** | 0.53 | 48 |
| | D-LO-P&X | $808{,}391 \pm 0$ | 808,391 | **26.92** | 2,669 | $807{,}673 \pm 2{,}155$ | **801,207** | **9.22** | 859 |
| | K-means | $919{,}058 \pm 73{,}594$ | 734,571 | 0.71 | 35 | $697{,}527 \pm 32{,}211$ | 643,583 | 0.48 | 23 |
| | C-LO | $919{,}058 \pm 73{,}594$ | 734,571 | 0.70 | 35 | $697{,}527 \pm 32{,}211$ | 643,583 | 0.46 | 23 |
| 10 | D-LO | $637{,}004 \pm 4{,}319$ | **625,281** | 19.52 | 870 | $634{,}216 \pm 5{,}596$ | 625,467 | 6.18 | 288 |
| | Min-D-LO | $642{,}980 \pm 7{,}605$ | 637,400 | 6.65 | 317 | $634{,}293 \pm 6{,}477$ | 625,468 | 2.55 | 125 |
| | D-LO-P&X | $\mathbf{636{,}937 \pm 6{,}825}$ | **625,281** | **145.20** | 7,310 | $\mathbf{632{,}331 \pm 5{,}715}$ | **623,701** | **40.77** | 2,084 |
| | K-means | $790{,}822 \pm 88{,}437$ | 650,038 | 1.65 | 34 | $529{,}028 \pm 32{,}301$ | 487,823 | 1.25 | 26 |
| | C-LO | $790{,}822 \pm 88{,}437$ | 650,038 | 1.67 | 34 | $529{,}028 \pm 32{,}301$ | 487,823 | 1.32 | 26 |
| 25 | D-LO | $481{,}983 \pm 5{,}198$ | 475,651 | 155.39 | 3,016 | $475{,}299 \pm 3{,}831$ | 468,201 | 35.96 | 705 |
| | Min-D-LO | $485{,}809 \pm 6{,}874$ | 473,159 | 40.17 | 787 | $\mathbf{474{,}431 \pm 4{,}508}$ | **467,745** | 15.77 | 316 |
| | D-LO-P&X | $\mathbf{480{,}415 \pm 6{,}816}$ | **468,503** | **631.20** | 12,799 | $475{,}962 \pm 4{,}071$ | 469,631 | **225.36** | 4,682 |
| | K-means | $731{,}980 \pm 91{,}535$ | 552,717 | 2.72 | 28 | $439{,}029 \pm 10{,}015$ | 418,754 | 3.02 | 31 |
| | C-LO | $731{,}980 \pm 91{,}535$ | 552,717 | 2.72 | 28 | $439{,}029 \pm 10{,}015$ | 418,754 | 2.96 | 31 |
| 50 | D-LO | $400{,}826 \pm 2{,}920$ | 395,833 | 314.36 | 3,133 | $\mathbf{392{,}016 \pm 1{,}513}$ | **388,746** | 157.97 | 1,228 |
| | Min-D-LO | $402{,}716 \pm 2{,}469$ | 398,453 | 103.30 | 1,040 | $392{,}146 \pm 2{,}080$ | 388,990 | 60.41 | 533 |
| | D-LO-P&X | $\mathbf{400{,}027 \pm 3{,}553}$ | **393,731** | **1,214.16** | 12,563 | $393{,}227 \pm 2{,}203$ | 390,815 | **596.58** | 6,220 |

*Table 11.* News20 dataset 2 ($N = 200, d = 130{,}107$): Mean, variance, and minimum of the clustering loss, along with the average computation time and average number of iterations over 20 runs for each initialization method and number of clusters, for K-means, C-LO, D-LO, Min-D-LO, and D-LO-P&X with $D$ chosen as the squared Euclidean distance.

| | Initialization | Random | | | | K-means++ | | | |
|---|---|---|---|---|---|---|---|---|---|
| $K$ | Algorithm | Mean ± Variance | Minimum | Time(s) | Num Iter | Mean ± Variance | Minimum | Time(s) | Num Iter |
| | K-means | $108{,}473 \pm 16{,}501$ | 97,481 | 1.80 | 13 | $104{,}391 \pm 17{,}022$ | 97,522 | 0.87 | 6 |
| | C-LO | $108{,}473 \pm 16{,}501$ | 97,481 | 1.77 | 13 | $104{,}391 \pm 17{,}022$ | 97,522 | 0.86 | 6 |
| 5 | D-LO | $\mathbf{97{,}473 \pm 3}$ | **97,472** | 3.20 | 21 | $\mathbf{97{,}944 \pm 2{,}046}$ | **97,472** | 2.67 | 17 |
| | Min-D-LO | $97{,}944 \pm 2{,}046$ | **97,472** | 3.46 | 21 | $99{,}134 \pm 3{,}323$ | **97,472** | 1.91 | 12 |
| | D-LO-P&X | $97{,}947 \pm 2{,}045$ | **97,472** | **90.99** | 622 | $97{,}947 \pm 2{,}045$ | **97,472** | **11.04** | 79 |
| | K-means | $91{,}420 \pm 1{,}988$ | 85,537 | 3.09 | 11 | $82{,}519 \pm 3{,}638$ | 76,319 | 2.07 | 7 |
| | C-LO | $91{,}420 \pm 1{,}988$ | 85,537 | 2.97 | 11 | $82{,}519 \pm 3{,}638$ | 76,319 | 2.08 | 7 |
| 10 | D-LO | $77{,}046 \pm 1{,}540$ | 74,083 | 44.97 | 144 | $75{,}730 \pm 1{,}500$ | 74,044 | 25.54 | 77 |
| | Min-D-LO | $76{,}064 \pm 915$ | 74,653 | 23.71 | 80 | $\mathbf{75{,}359 \pm 1{,}082}$ | 74,044 | 14.46 | 45 |
| | D-LO-P&X | $\mathbf{76{,}025 \pm 1{,}456}$ | **74,064** | **206.48** | 763 | $76{,}472 \pm 1{,}564$ | **73,840** | **51.92** | 181 |
| | K-means | $80{,}707 \pm 3{,}564$ | 73,203 | 5.43 | 9 | $61{,}430 \pm 3{,}570$ | 54,931 | 4.34 | 7 |
| | C-LO | $80{,}707 \pm 3{,}564$ | 73,203 | 5.37 | 9 | $61{,}430 \pm 3{,}570$ | 54,931 | 4.35 | 7 |
| 25 | D-LO | $\mathbf{50{,}773 \pm 642}$ | **49,960** | 317.12 | 464 | $\mathbf{50{,}220 \pm 443}$ | **49,503** | 144.74 | 210 |
| | Min-D-LO | $51{,}224 \pm 787$ | 50,,235 | 104.04 | 153 | $50{,}591 \pm 362$ | 49,820 | 61.32 | 88 |
| | D-LO-P&X | $51{,}018 \pm 756$ | 50,117 | **762.70** | 1,190 | $50{,}394 \pm 349$ | 49,575 | **228.14** | 353 |
| | K-means | $66{,}502 \pm 4{,}580$ | 58,491 | 9.66 | 8 | $41{,}926 \pm 3{,}762$ | 35,524 | 7.35 | 6 |
| | C-LO | $66{,}502 \pm 4{,}580$ | 58,491 | 9.71 | 8 | $41{,}926 \pm 3{,}762$ | 35,524 | 7.29 | 6 |
| 50 | D-LO | $\mathbf{32{,}108 \pm 134}$ | **31,923** | 1,014.97 | 769 | $\mathbf{31{,}945 \pm 102}$ | **31,819** | 470.17 | 356 |
| | Min-D-LO | $32{,}391 \pm 193$ | 32,032 | 298.15 | 227 | $32{,}205 \pm 182$ | 31,897 | 154.54 | 118 |
| | D-LO-P&X | $32{,}197 \pm 138$ | 31,995 | **1,684.39** | 1,322 | $32{,}006 \pm 97$ | 31,827 | **661.54** | 518 |

