# OpenReview forum: "Modified K-means Algorithm with Local Optimality Guarantees"
_ICML.cc/2025/Conference — ICML 2025 poster_

### Official Review · Reviewer_fHBw · 2025-03-09

**Overall Recommendation:** 3

**Summary:**

This paper generalizes necessary and sufficient conditions for local optimality of a solution of the continuous relaxation of the k-means problem; they generalize these conditions from the case using the Euclidean dissimilarity (Peng & Xia, 2005) to Bregman divergence. Similarly, they then use these observations to design extensions to the k-means algorithm that take effect when the k-means algorithm converges to a partition where a point is equidistant to at least two cluster centers. The returned partition is then a local optimum of the continuously relaxed k-means problem. Experiments show that their methods achieve better objective values in certain synthetic and real world settings.

**Claims And Evidence:**

Given the work of (Peng & Xia, 2005), I think the claim there is still a gap between convergence and local optimality is a little misleading. However, this is not widely known. Having skimmed that paper, this paper does a much better job of presenting the issue to readers not intimately family with optimization, in particular, cutting plane methods.

**Essential References Not Discussed:**

(Peng & Xia, 2005) is briefly mentioned but should have been discussed a lot more.

**Experimental Designs Or Analyses:**

No.

**Methods And Evaluation Criteria:**

The methods and evaluation criteria make sense.

**Other Comments Or Suggestions:**

Adding the number of iterations to Table 1 etc would be useful. I assume the time complexity of D-LO-K-means++ seems to be quadratic in the number of clusters because as k increases, the number of iterations increases linearly. Having the number of iterations would let me verify this. This is interesting in of itself since for kmeans++, the runtime appears linear in the number of clusters.

**Other Strengths And Weaknesses:**

The main strength of this paper is a much simpler algorithm for returning continuously/discretely locally optimal partitions compared to the cutting planes method of (Peng & Xia, 2005) as well as addressing the common mis-understanding that convergence implies local optimality.

The experimental results are promising but the runtimes seem very slow. I would have liked to have seen more results in a runtime constrained setting where each algorithm has a fixed time budget.

The main weakness is the similarity to (Peng & Xia, 2005) and the lack of discussion on this point.

**Questions For Authors:**

1) In section 5.3.2, the results suggest that the k-means algorithm seems to converge to C-local optimal solutions in most real-world datasets. Is there a way you could formalize this? Maybe through a smoothed analysis? This would be useful to know either way.

2) There are many techniques for speeding up k-means (minibatches/coresets/triangle inequality tricks etc), would any of them compose nicely with your methods? Either in theory or practice? This would make this work more significant as vanilla k-means is often considered to be too slow for massive datasets.

3) See previous questions regarding (Peng & Xia, 2005).

**Relation To Broader Scientific Literature:**

I think more time should have been devoted to distinguishing the results of this paper from that of  (Peng & Xia, 2005), both the theory and the algorithms. For example, is the only difference between Lemma 2.1 (Peng & Xia, 2005) and Theorem 4.2 of this paper, the difference in choice of dissimilarity, or does the proof use different techniques? Using the euclidean dissimilarity, how do the algorithms presented here compare to the cutting plane method from (Peng & Xia, 2005)? - I understand implementing that method may be a lot of work so even an intuitive understanding would be useful. For example, does the cutting plane method break down when the Bregman convergence is used?

**Theoretical Claims:**

I have not checked any of the proofs rigorously.

---

> ### Author Rebuttal · Authors · 2025-03-31
>
> We thank Reviewer fHBw for their review.
>
> **1. High-level comparison with Peng & Xia:**
>
> **R1:** Both of our works study the K-means problem, but our focus is on convergence of Lloyd's algorithm, whereas they consider completely different methods to obtain local minima, to ultimately find a global optimal solution. Given their focus on global optimization, we agree that perhaps our paper "does a much better job of presenting the issue", as in Section 4, as part of their experiments, it is stated that "we run the K-means algorithm to find a local optimum of (1.5)", which our paper clearly shows does not hold in general.
>
> Other methods exist to solve the K-means problem with varying convergence guarantees, but Lloyd's algorithm is the clear dominant method, so our focus was placed on studying and establishing the convergence of Lloyd's algorithm, in particular, to a local minimum for practical reasons. Minimizing a concave function over a polytope (as in the K-means problem) is NP-hard, and according to (Peng & Xia, Section 4: Numerical Experiments), there seems to be no guarantee that their method will not have to traverse all vertices (clusterings) in the worst case scenario, with their algorithm never obtaining a globally optimal solution in any of their experiments.
>
> **2. Lemma 2.1 and Theorem 4.2:**
>
> **R2:** Theorem 4.2 is not simply a theoretical observation, but played a crucial role in the design of our modification of Lloyd's algorithm and the proof of its convergence. For Theorem 4.2, much effort was placed on finding a minimum number of conditions which could be easily verified and implemented within Lloyd's algorithm, also without increasing its per iteration computational complexity. We do not want to downplay the fact that we consider Bregman divergences, but we believe the difference between our results run much deeper.
>
> **3. Empirical comparison with Peng & Xia:**
>
> **R3:** We compared our algorithms, with (Peng & Xia, Section 3.2.1)'s algorithm to find D-local optimal solutions, which is not based on Lloyd's algoirthm, but rather on repeatedly performing single-point swaps to cluster assignments, which can be found in Section 2 of https://anonymous.4open.science/r/ICML-Kmeans-F32E/Additional_Experiments.pdf, where we have also included the number of iterations for the K-means based algorithms. From these results, we observe that D-LO-K always acheives the minimum mean error (blue), whereas D-LO-P&X is consistently much slower than all other methods (red).
>
> **4. Runtime constrained experiments:**
>
> **R4:** Our response to Reviewer BPFU (R2 paragraph 2) establishes that our methods will always perform at least as well as Lloyd's method under an iteration or time constrained setting. In our experiments, we never had a maximum iteration stopping rule, simply letting our methods run until convergence. Inspired by your request, we plot the objective function through time for D-LO in Section 3 of our attachment, where in black it is identical to K-means, and in red is when after K-means has converged, it begins to call Function 2. From these plots, if we limit iterations to around 200, we can still achieve approximately 15% objective function value improvement with 3.5-5x algorithm speedup.
>
> **5. Theoretical understanding of K-means converging to C-local optimal solutions:**
>
> **R5:** Assume the data X is sampled from an absolutely continuous probability distribution, e.g. normally distributed, and D is the squared Euclidean distance. The cluster centers, being weighted averages of elements of X are also absolutely continuously distributed. For the K-means algorithm to not converge to a C-local optimum, it needs to converge to a clustering P where there exists a point x and two clusters c_1 and c_2 such that d(x,c_1)=d(x,c_2), where d is the Euclidean distance. This means that c_1 and c_2 need to lie on the surface of the same sphere, which has measure (probability) 0.
>
> **6. Composing with other techniques:**
>
> **R6:** Minibatch K-means is not a variant of Lloyd's algorithm, but of stochastic gradient descent, so our analysis is not applicable. From (Bottou & Bengio, 1994, Section 3.4), the method seems to converge to a local minimum almost surely. In practice, minibatch K-means can be faster but with a sacrifice in quality (Web-Scale K-Means Clustering, Sculley, 2010, Figure 1). Using a coreset $Y\subset X$ of size m<N instead of X in Lloyd's algorithm, our method can be applied to generate a locally optimal cluster C*, whose loss function is no worse than $(1+\epsilon)$ times the loss over $X$. The per-iteration complexity of Theorem 4.5 will be reduced from $O(Nkd)$ to $O(mkd)$ (for squared Euclidean norm), and will certainly result in faster compute time. Given that Elkan's method is still performing Lloyd's algorithm, but in a (computationally) optimized way by using the triangle inequality to reduce the number of distance computations, our method can also be used with Elkan's method.

---

### Official Review · Reviewer_bhiT · 2025-03-12

**Overall Recommendation:** 3

**Summary:**

This paper investigates the local optimality properties of the K-means clustering algorithm and proposes modifications that guarantee local optimality in both continuous and discrete senses. The authors introduce theoretical results that highlight scenarios where the standard K-means algorithm does not always converge to a local minimum and propose an improved version, LO-K-means, which achieves local optimality while maintaining the same computational complexity as K-means. The method is evaluated on synthetic and real-world datasets, demonstrating improved convergence properties.

**Claims And Evidence:**

yes

**Essential References Not Discussed:**

No

**Experimental Designs Or Analyses:**

The paper would benefit from including larger datasets, as suggested, to evaluate the scalability of LO-K-means.

**Methods And Evaluation Criteria:**

The choice of dataset sizes is relatively small. As suggested, larger datasets (N > 1000) should be considered to assess scalability.

**Other Comments Or Suggestions:**

Appendix C contains an incorrect reference: "(see Appendix C for full details)." This suggests a lack of careful proofreading. The authors should ensure consistency in citations and references.

**Other Strengths And Weaknesses:**

Strengths:
1. Strong theoretical contributions with clear mathematical formulations.

Weaknesses:
1. Larger datasets are needed to validate scalability.
2. Some theoretical claims (e.g., Equation 1 motivation) require better clarification.

**Questions For Authors:**

What is the primary motivation for Equation (1), and could it be presented in a more intuitive manner?

**Relation To Broader Scientific Literature:**

While the use of Bregman divergences is mentioned, a more detailed comparison with alternative clustering approaches (e.g., spectral clustering, Gaussian mixture models) would strengthen the discussion.

**Theoretical Claims:**

The notation and presentation in some places (e.g., Appendix C) could be more refined for clarity.

---

> ### Author Rebuttal · Authors · 2025-03-30
>
> We thank Reviewer bhiT for their suggestions and the positive feedback, that our work has "strong theoretical contributions with clear mathematical formulations".
>
> **1. Test on larger datasets (N > 1000):**
>
> **R1:** We want to highlight that several experiments were done for datasets with N>1000: Wine Quality (N=6,497), Yeast (N=1,484), Predict Students’ Dropout and Academic Success (N=4,424), and News20 (with N=2,000). We refer Reviewer bhiT to Appendix E.2, where all of the details of our real-world datasets are contained. Given that these are all real-world datasets, we thought that perhaps the reviewer desired experiments also using large-sample synthetic datasets, as our synthetic dataset experiments were only initially done for N up to 300. Additional large-sample experiments on synthetic datasets can be found in Section 1 of https://anonymous.4open.science/r/ICML-Kmeans-F32E/Additional_Experiments.pdf. These experiments were done with datasets of up to 20,000 samples. Given that our C-local method produces more modest improvements compared to our D-local method, we did these experiments with C-LO-K-means, with the K-means++ initialization. We observe that even with these large-sample datasets, C-LO-K-means can outperform the standard K-means algorithm.
>
> **2. Comparing with alternative clustering approaches (e.g., spectral clustering, Gaussian mixture models):**
>
> **R2:** This work is focused on improving Lloyd's algorithm. We attempted to consider a general setting (weighted K-means using Bregman divergences), but unfortunately not all clustering methods could be considered in this first paper. We do hope to generalize our results though to other clustering problems in future work.
>
> Spectral clustering for graphs, which clusters data points based on their connectivity, can be accomplished by clustering the eigenvectors of the K smallest eigenvalues of the Laplacian using the K-means algorithm, so for this problem, our method can be applied to improve the clustering stage.
>
> There are similarities between the K-means algorithm and the EM algorithm, used for estimating Gaussian mixture models, i.e., in the E step, data points are assigned to clusters, and the M step computes the cluster centers. The possible extension of our work to EM algorithms naturally interests us. Knowledge that the data points are sampled from a mixture of Gaussian distributions is fully exploited in the EM algorithm to estimate and maximize the model parameters' log-likelihood function, whereas Lloyd's algorithm is much simpler (e.g. no covariance estimation). Our method is not attempting to change this about Lloyd's algorithm, so we can still expect its performance on GMM to be better than Lloyd's algorithm, but not to be able to compare to the EM algorithm in terms of the complexity of clusters that it can generate.
>
> **3. Equation 1 motivation:**
>
> **R3:** Given that the motivation for this work is about the local optimality of Lloyd's algorithm, it was most convenient to formulate the K-means problem as a mathematical optimization problem for our analysis. Similar formulations can be found, for example, in
> (Selim and Ismail, K-Means-Type Algorithms: A Generalized Convergence Theorem and Characterization of Local Optimality, 1984, Equation 1) and (Peng and Xia, A Cutting Algorithm for the Minimum Sum-of-Squared Error Clustering, 2005, Equation 1.2). Given that we study both continuous and discrete local optimality, we wanted to clearly isolate the difference between (P1) and its continuous relaxation (P2), which motivated the separation of constraints into sets S1 in (P1) and S2 in (P2). Given that we consider general Bregman divergences, we also needed to consider the domain of the cluster centers, written as $R$, which is no longer always simply $\mathbb{R}^d$ as is the case with the squared Euclidean distance.
>
> **4. Appendix C reference:**
>
> **R4:** We apologize for the confusion regarding "(see Appendix C for full details)" in Appendix C. In the body we tried to include as much of our full counterexample as possible, which is contained in the appendix, so this exact paragraph is contained in both places. We also noticed this, and removed it from our current version of the paper. Given that our main focus was on the technical correctness of our claims, this type of silly mistake was able to sneak through.
>
> We hope that we have properly answered all of your questions, and that in particular, you are satisfied with our use of N>1000-sample datasets in our experiments. We would greatly appreciate it if you would be able to consider increasing your overall recommendation score.

---

### Official Review · Reviewer_A9GB · 2025-03-13

**Overall Recommendation:** 2

**Summary:**

This paper considers a (natural) notion of local-optimality for the k-means problem, and shows that Lloyd's algorithm can lead to solutions that are not locally optimal. Generally when anyone discusses Lloyd's algorithm, they often claim that Lloyd's gets stuck in a "local minima" so this result is interesting. The paper further shows the necessary and sufficient conditions for a solution to be locally optimal and proposes a simple modification of Lloyd's algorithm, basically by augmenting it with a local-search step at the end, to provide an algorithm with guaranteed local minima.

**Claims And Evidence:**

Yes

**Essential References Not Discussed:**

There is a HUGE swath of work related to Local search algorithms for the k-means problem, starting with the paper *A local search approximation algorithm for k-means clustering* by Kanungo et.al (2004).  Given that the paper (essentially) proposes a local-search to be done after Lloyd's iterations converge, it makes sense that the paper at least discusses how prior works on Local Search for k-means are related to this. Honestly, it felt like this paper could have come out in the 1980's (soon after the Selim and Ismail 1984 paper) and there's no reason to believe that it would have been very different...

**Experimental Designs Or Analyses:**

Yes

**Methods And Evaluation Criteria:**

Yes

**Other Comments Or Suggestions:**

Line 25 on the right hand paragraph should be *Grunau and Rozhon confirm that* instead of *Gruanu and Bock confirm that*

**Other Strengths And Weaknesses:**

Yes, the contributions are somewhat interesting but I feel it falls short of an ICML 2025 paper...

**Questions For Authors:**

None

**Relation To Broader Scientific Literature:**

Yes, the paper's contributions are related to the scientific literature. At this point, Lloyd's algorithm is so commonly encountered that giving a link to a single (or a few) specific paper is not required.

**Theoretical Claims:**

I went over the proofs and they mostly seem correct. It's unlikely there is any major flaw in the claims...

---

> ### Author Rebuttal · Authors · 2025-03-29
>
> We thank the reviewer for their comments and suggestions, and are happy that they found our result to be interesting.
>
> **1. Local search algorithms:**
>
> **R1:** Similar to K-means++,  Kanungo et. al present a heuristic to initialize centers, with a guarantee that the objective value (distortion) of their initialization is no greater than 9+epsilon times worse than the optimal objective function value. After using their local-search initialization, they suggest to use Lloyd's algorithm.
>
> This work is complementary to ours, as it is focused on how to start solving the K-means problem, whereas our work can be viewed as how to finish solving the K-means problem. Our work is not dependent on how the K-means solution is initialized
> so, following scikit-learn, we considered both K-means++ and randomly choosing data points as centers, which seem to be the most popular choices. K-means++ also bounds its initialization, though in expectation and as a function of K. How to initialize centers is adjacent to our work, though given the natural trade-off between speed and accuracy, the simplicity of K-means++ likely plays a major role in its popularity, as it is very easy to implement, whereas in Kanungo et. al's implementation of their method in their Experimental Results section, they were required to make simplifications to their method, seemingly losing its theoretical guarantees.
>
> In particular, Kanungo et. al's method defines a large set of candidate centers C such that it contains centers which can form an $\epsilon$-approximation. Initializaing S as a random sampling of K cluster centers from C, their approach consists of randomly swapping out clusters from S with clusters from C and seeing if the distortion is improved. This work presents their results in terms of "stability", but this should not be confused with some notion of the local optimality of the K-means problem, as it is in reference to only their initialization heuristic and the swapping of candidate cluster centers.
>
> Our method on the other hand is not a heuristic. The "search" aspect of our method, if we want to call it that, deterministically verifies if the condition for local optimality of the K-means problem holds after Lloyd's algorithm has terminated. If we find that it has not converged to a local minimum, meaning that we have found that the objective function is guaranteed to strictly decrease by moving a point to a different cluster, we do this operation, and then continue running Lloyd's algorithm. In Kanungo et.al, they assume that "Lloyd’s algorithm eventually converges to a locally optimal solution", so our method can be used to guarantee that this holds, perhaps strengthening the arguments in their work.
>
> **2. 1980’s paper:**
>
> **R2:** We agree, in that it is remarkable that after over 40 years we are able to point out an error in the convergence analysis of the K-means algorithm and properly address the non-convergence of this method. We think that our work is important for the community to show that these types of ``folklore" results cannot always be blindly trusted.
>
> **3. Grunau and Rozhon:**
>
> **R3:** Thank you for pointing that out. We have made the correction.
>
> **4. Falls short of an ICML 2025 paper:**
>
> **R4:** Our work brings a clear and deeper understanding of the K-means algorithm, and presents a simple method to improve its solution while guaranteeing local optimality. Given the large number of methods which use Lloyd's algorithm as a base to solve the K-means problem, and it being itself "by far the most popular clustering algorithm used in scientific and industrial applications” (Pavel Berkhin. Survey of clustering data mining techniques, 2002), we believe our work is important and has significant impact. We would be happy to answer any further questions, but if we have satisfied your concerns we would appreciate it if you would consider increasing your overall recommendation of 2.

---

### Official Review · Reviewer_BPFU · 2025-03-13

**Overall Recommendation:** 3

**Summary:**

The paper shows that the traditional K-means algorithm does not always converge to a local optimum (by a 1D counterexample). The paper proves the conditions for K-means to converge to a local optimum. By modifying the termination conditions of K-means (adding a new step), we can guarantee convergence to either a continuous (C-local) or discrete (D-local) local optimum.

**Claims And Evidence:**

The claims are well-supported by theoretical analysis and experiments.

**Essential References Not Discussed:**

None. It is just a simple modification of the K-means algorithm.

**Experimental Designs Or Analyses:**

Overall, the experiments covered all the theoretical claims. I'm not very familiar with the more detailed experimental design so I can't comment. I haven't tried to reproduce the result.

**Methods And Evaluation Criteria:**

This paper is a simple modification based on existing algorithms (K-means, K-means++). The experiments were tested on synthetic and real-world datasets. Overall, the method and evaluation both make sense.

**Other Comments Or Suggestions:**

Overall, I believe this is a solid contribution.

**Other Strengths And Weaknesses:**

Strengths:

-very natural setting, and very natural optimization question for k-means.

-This paper gives the first rigorous disproof of K-means’ local optimality and a practical fix.

-The algorithms and conclusions are simple and elegant.

One possible weakness would be the improvement in C-LO is not significant, while D-LO adds a considerable computational overhead.

**Questions For Authors:**

One quick question: Isn't the modification essentially a tie-breaker? Could randomly selecting neighbors/SGD also make the K-means algorithm converge to the local optimum with high probability? (Your algorithm is deterministic, which is an advantage. It would be better if there is a discussion on randomness.) Have people studied this version and what is it known in the literature? It seems like a simple modification that people must have studied? Thanks!

**Relation To Broader Scientific Literature:**

K-means has some variants, such as X-means and G-means:

Pelleg, Dan, and Andrew Moore. "X-means: Extending K-means with Efficient Estimation of the Number of Clusters." ICML’00. Citeseer, 2000.

Hamerly, Greg, and Charles Elkan. "Learning the k in k-means." Advances in neural information processing systems 16 (2003).

This remains an open question: Will they converge to a local optimum? Are there similar improvements? Can the authors comment from their experience on the performance of such algorithms?

**Theoretical Claims:**

I checked the proof and it looks solid.

---

> ### Author Rebuttal · Authors · 2025-03-29
>
> We thank the reviewer for their thoughtful review and positive comments, namely that "the claims are well-supported by theoretical analysis and experiments", "the algorithms and conclusions are simple and elegant", and that "overall...this is a solid contribution".
>
> **1. X-means & G-means:**
>
> **R1:** In our work, which considers the classic K-means problem, we assume that K is given. X-means tries to find the optimal K based on the Bayesian information criterion (BIC). The goal of their work is somewhat adjacent to ours, but we verified that our algorithm can be used within their method: Step 1 (Improve-Params) consists of running conventional K-means to convergence, so this can be improved using our method. Step 2 (Improve-Structure) splits clusters in two, does local 2-means for each pair of clusters, which again can use our method, and then uses BIC to determine whether to keep the 2 children clusters or the original cluster. This work is using a heuristic method, searching for the best K over a given range, based on a statistical criterion. We note that in terms of minimizing the clustering error, one would always choose the largest possible K, so this work does not present any type of "local optimality guarantee" for the choice of K from an optimization perspective, nor for the K-means problem for the chosen K.
>
> As described in their paper, G-means is another "wrapper around K-means", trying to find the appropriate choice of K by running the K-means algorithm for consecutively higher K until a statistical test is satisfied. Somewhat similarly to X-means, clusters are split into two, with their centers computed using K-means, so our K-means algorithm can also be applied within the G-means algorithm to improve the accuracy of their method.
>
> G-means tests if the within cluster data is sufficiently Gaussian, hence it is highly dependent on the squared Euclidean loss, where our work does not rely on any assumptions on the distribution of the underlying data, while considering general Bregman divergences. Similarly X-means, using BIC, requires the log-likelihood of the data, which they calculate assuming the data is spherical Gaussian.
>
> **2. Improvement using C-LO & computation overhead of D-LO:**
>
> **R2:** Up until now there was no simple method to verify the quality of the solution of Lloyd's algorithm. Generating a C-local minimum is fast, and if Lloyd's algorithm has converged to a C-local minimum, it only requires a single call to Function 1 to guarantee it. D-local is a stronger notion of optimality than C-local (Proposition 2.5), but naturally it is generally slower.
>
> The only difference between our methods and Lloyd's algorithm occurs after Lloyd's algorithm has converged. If Lloyd's algorithm does not converge to a local minimum, our methods will perform additional iterations which are all guaranteed to strictly decrease the objective function. Therefore, D-local's potentially slow performance can be controlled by setting a maximum iteration limit. With any fixed time or iteration budget, our methods will perform as well as Lloyd's algorithm: If Lloyd's algorithm converges within budget, our methods will output a solution with a lower objective function if they can perform addition iterations, or else our methods' solutions will match Lloyd's. If Lloyd's does not converge in time, our methods' solutions will again exactly match Lloyd's algorithm.
>
> In order to directly improve the computational overhead of D-LO, we developed an Accelerated D-LO algorithm (Accel-D-LO), see Section 3 of https://anonymous.4open.science/r/ICML-Kmeans-F32E/Additional_Experiments.pdf, where this new heuristic is tested, with a demonstration of the previous paragraph's message. When running D-LO-K-means, instead of simply choosing the first value of $n$ and $k_2$ such that $\Delta_1(n,k_1,k_2)<0$ in Function 2, Accel-D-LO finds the $n$ and $k_2$ which minimize $\Delta_1(n,k_1,k_2)$, moving the cluster assignment to the adjacent vertex that  decreases the objective function value the most. We observe that this simple heuristic speeds up D-LO-K-means 2-3X while still guaranteeing convergence to a D-local minimum.
>
> Since our method is a slight modification of the K-means algorithm, we also direct Reviewer BPFU to our rebuttal for Reviewer fHBw, where we discuss how our method is also compatible with techniques that can speed up the K-means algorithm such as using coresets and Elkan's method.
>
> **3. Randomly selecting neighbors/SGD for tie-breaking:**
>
> **R3:** Our focus on a deterministic rule was to present the simplest algorithm with our desired theoretical guarantees, but yes, finding all (or a subset) of tie-breakers and then randomly selecting one would still maintain our convergence guarantees. In our initial algorithms, we simply used the first tie-breaker that we found. We refer Reviewer BPFU to our rebuttal for Reviewer fHBw where we discuss minibatch K-means, which is an SGD-type method for the K-means problem.

---

### Decision · Program_Chairs · 2025-05-01

**Decision:**

Accept (poster)

**Comment:**

The paper offers insights into the local optimality of the k-means algorithm and proposes modifications to Lloyd’s algorithm targeting two conditions. Most reviewers agree that the paper is well-written and provides interesting insights, particularly the counterexample. Therefore, I recommend a (weak) accept.